# Minimax Optimality (Probably) Doesn't Imply Distribution Learning for GANs

**Sitan Chen**
UC Berkeley

**Jerry Li**
Microsoft Research

**Yuanzhi Li**
CMU

**Raghu Meka**
UCLA

## Abstract

Arguably the most fundamental question in the theory of generative adversarial networks (GANs) is to understand when GANs can actually learn the underlying distribution. Theoretical and empirical evidence (see e.g. (Arora et al., 2018)) suggests local optimality of the empirical training objective is insufficient. Yet, it does not rule out the possibility that achieving a true population minimax optimal solution might imply distribution learning. In this paper, we show that standard cryptographic assumptions imply this stronger condition is still insufficient. Namely, we show that if local pseudorandom generators (PRGs) exist, then for a large family of natural target distributions, there are ReLU network generators of constant depth and poly size which take Gaussian random seeds so that (i) the output is far in Wasserstein distance from the target distribution, but (ii) no polynomially large Lipschitz discriminator ReLU network can detect this. This implies that even achieving a population minimax optimal solution to the Wasserstein GAN objective is likely insufficient for distribution learning in the usual statistical sense. Our techniques reveal a deep connection between GANs and PRGs, which we believe will lead to further insights into the computational landscape of GANs.

## 1 Introduction

When will a generative adversarial network (GAN) trained with samples from a distribution $\mathcal{D}$ actually output samples from a distribution that is close to $\mathcal{D}$? This question is one of the most foundational questions in GAN theory—indeed, it was raised since the original paper introducing GANs. However, despite significant interest, this question still remains to be fully understood for general classes of generators and discriminators.

A significant literature has developed discussing the role of the training dynamics (Liu et al., 2017; Li et al., 2018; Arora et al., 2018; Berard et al., 2019; Wiatrak et al., 2019; Thanh-Tung & Tran, 2020; Allen-Zhu & Li, 2021), as well as the generalization error of the GAN objective (Zhang et al., 2017; Arora et al., 2017; Thanh-Tung et al., 2019). In most cases, researchers have demonstrated that given sufficient training data, GANs are able to learn some specific form of distributions after successful training. Underlying these works appears to be a tacit belief that if we are able to achieve the minimax optimal solution to the population-level GAN objective, then the GAN should be able to learn the target distribution. In this work, we take a closer look at this assumption.

**What does it mean to learn the target distribution?** As a starting point, we must first formally define what we mean by learning a distribution; more concretely, what do we mean when we say that two distributions are close? The original paper of (Goodfellow et al., 2020) proposed to measure closeness with KL divergence. However, learning the target distribution in KL divergence is quite unlikely to be satisfied for real-world distributions. This is because learning distributions in KL divergence also requires us to exactly recover the support of the target distribution, which we cannot really hope to do if the distribution lies in an unknown (complicated) low-dimensional manifold. To rectify this, one may instead consider learning in Wasserstein distance, as introduced in the context of GANs by (Arjovsky et al., 2017), which has no such "trivial" barriers. Recall that the Wasserstein distance between two distributions $\mathcal{D}_1, \mathcal{D}_2$ over $\mathbb{R}^d$ is given by

$$W_1(\mathcal{D}_1, \mathcal{D}_2) = \sup_{\text{Lip}(f) \leq 1} \mathbb{E}_{\mathcal{D}_1}[f] - \mathbb{E}_{\mathcal{D}_2}[f] \,,$$

where for any $f : \mathbb{R}^d \to R$, we let $\mathrm{Lip}(f)$ denote the Lipschitz constant of $f$. That is, two densities are close in Wasserstein distance if no Lipschitz function can distinguish between them. In this work we will focus on Wasserstein distance as it is the most standard notion of distance between probability distributions considered in the context of GANs.

Note that if the class of discriminators contains sufficiently large neural networks, then minimax optimality of the GAN objective does imply learning in Wasserstein distance. This is because we can approximate any Lipschitz function arbitrarily well, with an exponentially large network with one hidden layer (see e.g. (Poggio et al., 2017)). Thus, in this case, minimizing the population GAN objective is actually equivalent to learning in Wasserstein distance. Of course in practice, however, we are limited to polynomially large networks for both the generator and the discriminator. This raises the natural question:

> *Does achieving small error against all poly-size neural network discriminators imply that the poly-size generator has learned the distribution in Wasserstein distance?*

One might conjecture that this claim is true, since the generator is only of poly-size. Thus, using a (larger) poly-size discriminator (as opposite to the class of all 1-Lipschitz functions) might still be sufficient to minimize the actual Wasserstein distance. In this paper, however, we provide strong evidence to the contrary. We demonstrate that widely accepted cryptographic assumptions imply that this is false, *even if the generator is of constant depth*:

**Theorem 1.1** (Informal, see Theorem 3.1). *For any $n \in \mathbb{N}$, let $\gamma_n$ be the standard Gaussian measure over $\mathbb{R}^n$. Assuming local pseudorandom generators exist, the following holds for any sufficiently large $m \in \mathbb{Z}$, $d, r \leq \mathrm{poly}(m)$, and any diverse[1] target distribution $\mathcal{D}^*$ over $[0, 1]^d$ given by the pushforward of the uniform distribution $I_r$ on $[0, 1]^r$ by a constant depth ReLU network of polynomial size/Lipschitzness:[2]*

*There exist generators $G : \mathbb{R}^m \to \mathbb{R}^d$ computed by (deterministic) ReLU networks of* constant depth *and polynomial size for which no ReLU network of polynomial depth/size/Lipschitzness can tell apart the distributions $G(\gamma_m)$ and $\mathcal{D}^*$, yet $G(\gamma_m)$ and $\mathcal{D}^*$ are $\Omega(1)$-far in Wasserstein distance.*

While Theorem 1.1 pertains to the practically relevant setting of *continuous* seed and output distributions, we also give guarantees for the discrete setting. In fact, if we replace $\mathcal{D}^*$ and $\gamma_m$ by the uniform distributions over $\{\pm 1\}^d$ and $\{\pm 1\}^m$, we show this holds for generators whose output coordinates are given by *constant-size* networks (see Theorem 3.2).

We defer the formal definition of local pseudorandom generators (PRGs) to Section 2.2. We pause to make a number of remarks about this theorem.

First, our theorem talks about the population loss of the GAN objective; namely, it says that the true population GAN objective is small for this generator $G$, meaning that for every ReLU network discriminator $f$ of polynomial depth/size/Lipschitzness, we have that

$$|\mathbb{E}[f(\mathcal{D}^*)] - \mathbb{E}[f(G(\gamma_m))]| \leq \frac{1}{d^{\omega(1)}} \ .$$

In other words, our theorem states that even optimizing the true population minimax objective is insufficient for distribution learning. In fact, we show this even when the target distribution can be represented *perfectly* by some other generative model.

Second, notice that our generator is extremely simple: notably, it is only constant depth. On the other hand, the discriminator is allowed to be much more complex, namely any ReLU network of polynomial complexity. This discriminator class thus constitutes the most powerful family of functions we could hope to use in practice. Despite this, we show that the discriminators are still not powerful enough to distinguish the output of the (much simpler) generator from the target distribution.

Third, our conclusions hold both for $d \geq m$ and $d \leq m$, so long as the input and output dimensions are related by polynomial factors.

---

[1]See Definition 6. In the discussion proceeding this definition, we give a number of examples making clear that this is a mild and practically relevant assumption to make.

[2]When we say "polynomial," we are implicitly referring to the dependence on the parameter $m$, though because $d, r$ are bounded by $\mathrm{poly}(m)$, "polynomial" could equivalently refer to the dependence on those parameters if they exceeded $m$.

Finally, we formally define the class of "diverse" target distributions for which our conclusions hold in Section 2.3. We note that this class is quite general: for instance, it includes pushforwards of the uniform distribution under random leaky ReLU networks (see Lemma 2.3).

**Empirical results.** To complement these theoretical results, we also perform some empirical validations of our findings (see Section 4). Our theorem is constructive; that is, given a local PRG, we give an explicit generator which satisfies the theorem. We instantiate this construction with Goldreich's PRG with the "Tri-Sum-And" (TSA) predicate (Goldreich, 2011), which is an explicit function which is believed to satisfy the local PRG property. We then demonstrate that a neural network discriminator trained via standard methods empirically cannot distinguish between the output of this generator and the uniform distribution. While of course we cannot guarantee that we achieve the truly optimal discriminator using these methods, this still demonstrates that our construction leads to a function which does appear to be hard to distinguish in practice.

**GANs, PRGs, and circuit lower bounds.** At a high level, our results and techniques demonstrate surprising and deep connections between GANs and more "classical" problems in cryptography and complexity theory. Theorem 1.1 already shows that cryptographic assumptions may pose a fundamental barrier to the most basic question in GAN theory. In addition to this, we also show a connection between this question and circuit lower bounds. In the supplementary material, we show that if we are able to unconditionally exhibit generators which can fool polynomially large ReLU network discriminators, then we would obtain breakthrough circuit lower bounds against $TC^0$ (see Theorem C.2 and Remark C.3). This complements Theorem 1.1, as it says that if we can unconditionally construct generators which fool realistic discriminators, then we make progress on long-standing questions in circuit complexity. We believe that exploring these connections may be crucial to achieving a deeper understanding of what GANs can and cannot learn.

## 1.1 RELATED WORK

**GANs and Distribution Learning**    The literature on GAN theory is vast and we cannot hope to do it full justice here. For a more extensive review, see e.g. (Gui et al., 2020). Besides the previously mentioned work on understanding GAN dynamics and generalization, we only mention the most relevant papers here. One closely related line of work derives concrete bounds on when minimax optimality of the GAN objective implies distribution learning (Bai et al., 2018; Liang, 2018; Singh et al., 2018; Uppal et al., 2019; Chen et al., 2020a; Schreuder et al., 2021). However, the rates they achieve scale poorly with the dimensionality of the data, and/or require strong assumptions on the class of generators and discriminators, such as invertability. Another line of work has demonstrated that first order methods can learn very simple GAN architectures in polynomial time (Feizi et al., 2017; Daskalakis et al., 2017; Gidel et al., 2019; Lei et al., 2020). However, these results do not cover many of the generators used in practice, such as ReLU networks with $> 1$ hidden layers.

**Local PRGs and Learning**    PRGs have had a rich history of study in cryptography and complexity theory (see e.g. (Vadhan, 2012)). From this literature, the object most relevant to the present work is the notion of a *local PRG*. These are part of a broader research program of building constant parallel-time cryptography (Applebaum et al., 2006). One popular local PRG candidate was suggested in (Goldreich, 2011). By now there is compelling evidence that this candidate is a valid PRG, as a rich family of algorithms including sum-of-squares (ODonnell & Witmer, 2014) and statistical query algorithms (Feldman et al., 2018) provably cannot break it.

Finally, we remark that Goldreich's PRG and, more generally, hardness of refuting random CSPs have been used in a number of works showing hardness for various supervised learning problems (Daniely & Vardi, 2021; Daniely et al., 2014; Daniely, 2016; Daniely & Shalev-Shwartz, 2016; Applebaum et al., 2006; Applebaum & Raykov, 2016). We consider a very different setting, and our techniques are very different from the aforementioned papers.

## 2 TECHNICAL PRELIMINARIES

**Notation**    Denote by $U_n$ the uniform distribution over $\{\pm 1\}^n$, by $\gamma_n$ the standard $n$-dimensional Gaussian measure, and by $I_n$ the uniform measure on $[0, 1]^n$. Given distribution $\mathcal{D}$ over $\mathbb{R}^m$ and measurable function $G : \mathbb{R}^m \to \mathbb{R}^d$, let $G(\mathcal{D})$ denote the distribution over $\mathbb{R}^d$ given by the *push-*

*forward* of $\mathcal{D}$ under $G$—to sample from the pushforward $G(\mathcal{D})$, sample $x$ from $\mathcal{D}$ and output $G(x)$. Given $\sigma : \mathbb{R} \to \mathbb{R}$ and vector $v \in \mathbb{R}^d$, denote by $\sigma(v) \in \mathbb{R}^d$ the result of applying $\sigma$ entrywise to $v$. To avoid dealing with issues of real-valued computation, let $\mathbb{R}_\tau \subset \mathbb{R}$ be the set of multiples of $2^{-\tau}$ bounded in magnitude by $2^\tau$.

## 2.1 GANs and Pseudorandom Generators

In this section we review basic notions about generative models and PRGs.

**Definition 1** (ReLU Networks). *Let $\mathcal{C}_{L,S,d}$ denote the family of ReLU networks $F : \mathbb{R}^d \to \mathbb{R}$ of depth $L$ and size $S$. Formally, $F \in \mathcal{C}_{L,S,d}$ if there exist weight matrices $\mathbf{W}_1 \in \mathbb{R}^{k_1 \times d}, \mathbf{W}_2 \in \mathbb{R}^{k_2 \times k_1}, \ldots, \mathbf{W}_L \in \mathbb{R}^{1 \times k_{L-1}}$ and biases $b_1 \in \mathbb{R}^{k_1}, b_2 \in \mathbb{R}^{k_2} \ldots, b_L \in \mathbb{R}$ such that*

$$F(x) \triangleq \mathbf{W}_L \phi \left( \mathbf{W}_{L-1} \phi \left( \cdots \phi(\mathbf{W}_1 x + b_1) \cdots \right) + b_{L-1} \right) + b_L,$$

*and $\sum_{i=1}^{L-1} k_i = S$, where $\phi(z) \triangleq \max(0, z)$ is the ReLU activation. Additionally, we let $\mathcal{C}_{L,S,d}^{\tau,\Lambda}$ be the subset of such networks which are additionally $\Lambda$-Lipschitz and whose weight matrices and biases have entries in $\mathbb{R}_\tau$– we will refer to $\tau$ as the* bit complexity *of the network.*

The following allows us to control the complexity of compositions of ReLU networks:

**Lemma 2.1.** *Let $J : \mathbb{R}^s \to \mathbb{R}^r$ be a function each of whose output coordinates is computed by some network in $\mathcal{C}_{L_1,S_1,s}^{\tau_1,\Lambda_1}$, and let $f \in \mathcal{C}_{L_2,S_2,r}^{\tau_2,\Lambda_2}$. Then $f \circ J \in \mathcal{C}_{L,S,s}^{\tau,\Lambda}$ for $\tau = \max(\tau_1, \tau_2)$, $\Lambda = \Lambda_1 \Lambda_2 \sqrt{r}$, $L = L_1 + L_2$, and $S = (S_1 + 1)r + S_2$. Furthermore, for the network in $\mathcal{C}_{L,S,s}^{\tau,\Lambda}$ realizing $f \circ J$, the bias and weight vector entries in the output layer lie in $\mathbb{R}_{\tau_2}$.*

Next, we formalize the probability metric we will work with.

**Definition 2** (IPM). *Given a family $\mathcal{F}$ of functions, define the $\mathcal{F}$-integral probability metric between two distributions $p, q$ by $W_\mathcal{F}(p, q) = \sup_{f \in \mathcal{F}} |\mathbb{E}_{y \sim p}[f(y)] - \mathbb{E}_{y \sim q}[f(y)]|$. When $\mathcal{F}$ consists of the family of 1-Lipschitz functions, this is the standard* Wasserstein-1 *metric, which we denote by $W_1$.*

In the context of GANs, we will focus on discriminators given by ReLU networks of polynomial size, depth, Lipschitzness, and bit complexity:

**Definition 3** (Discriminators). *$\mathcal{F}^*$ denotes the set of all sequences of discriminators $f_d : \mathbb{R}^d \to \mathbb{R}$, indexed by $d \in \mathbb{N}$, whose size, depth, Lipschitzness, bit complexity grow at most polynomially in $d$.*

We now formalize the definition of GANs, which closely parallels the definition of PRGs.

**Definition 4** (GANs/PRGs). *Let $\epsilon : \mathbb{N} \to [0, 1]$ be an arbitrary function, and let $\{d(m)\}_{m \in \mathbb{N}}$ be some sequence of positive integers. Given a sequence of* seed *distributions $\{\mathcal{D}_m\}_m$ over $\mathbb{R}^m$, a sequence of* target *distributions $\{\mathcal{D}_{d(m)}^*\}$ over $\mathbb{R}^{d(m)}$, a family $\mathcal{F}$ of* discriminators *$f : \mathbb{R}^{d(m)} \to \mathbb{R}$, and a sequence of* generators *$G_m : \mathbb{R}^m \to \mathbb{R}^{d(m)}$, we say that $\{G_m\}$ $\epsilon$-fools $\mathcal{F}$ relative to $\{\mathcal{D}_{d(m)}^*\}$ with seed $\{\mathcal{D}_m\}$ if for all sufficiently large $m$,*

$$\left| \mathbb{E}[f(G_m(\mathcal{D}_m))] - \mathbb{E}[f(\mathcal{D}_{d(m)}^*)] \right| \leq \epsilon(m) \quad \forall f \in \mathcal{F}, f : \mathbb{R}^{d(m)} \to \mathbb{R}. \tag{1}$$

*Note that in the notation of Definition 2, (1) is equivalent to $W_\mathcal{F}(G_m(\mathcal{D}_m), \mathcal{D}_{d(m)}^*) \leq \epsilon(m)$.[3]*

*In this definition, if the discriminators and generators were instead Boolean functions, we would refer to $\{G_m\}$ as* pseudorandom generators.

*Remark* 2.2. It will often be cumbersome to refer to sequences of target/seed distributions and discriminators/generators as in Definitions 3 and 4, so occasionally we will refer to a single choice of $m$ and $d$ even though we implicitly mean that $m$ and $d$ are parameters that increase towards infinity. In this vein, we will often say that a single network $f$ is in $\mathcal{F}^*$, though we really mean that $f$ belongs to a sequence of networks which lies in $\mathcal{F}^*$. And for distributions $p, q$ which implicitly belong to sequences $\{p_d\}, \{q_d\}$, when we refer to bounds on $W_{\mathcal{F}^*}(p, q)$ we really mean that for any sequence of discriminators $f_d \in \mathcal{F}^*$, $|\mathbb{E}[f_d(p_d)] - \mathbb{E}[f_d(q_d)]|$ is bounded.

---

[3]Here we are slightly abusing notation as $\mathcal{F}$ contains functions which have domain not equal to $\mathbb{R}^{d(m)}$, and we ignore these functions in the supremum.

## 2.2 LOCAL PSEUDORANDOM GENERATORS

In complexity theory, the role of neural network discriminators is played by *Boolean circuits*, which one can roughly think of as networks which take in Boolean strings as input and whose activations are logical operations (e.g. AND/OR/NOT) rather than ReLUs (see Section A.2 in the supplement).

It is widely believed that there exist so-called *local PRGs* capable of fooling all polynomial-sized Boolean circuits and all of whose output coordinates are functions of a constant number of input coordinates (Applebaum et al., 2006). One prominent candidate is Goldreich's PRG:

**Definition 5** ((Goldreich, 2011)). *Let $H$ be a collection of $d$ subsets $S_1, \ldots, S_d$ of $\{1, \ldots, m\}$, each of size $k$ and each sampled independently from the uniform distribution over subsets of $\{1, \ldots, m\}$ of size $k$. Let $P : \{\pm 1\}^k \to \{\pm 1\}$ be a Boolean function; $P$ is often referred to as a* predicate.

*Let $G_{P,H} : \{\pm 1\}^m \to \{\pm 1\}^d$ denote the Boolean function whose $\ell$-th output coordinate is computed by evaluating $P$ on the coordinates of the input indexed by subset $S_\ell$ in $H$.*

The following is a standard assumption in cryptography (see (Applebaum, 2016)), namely that the incredibly simple functions in Definition 5 fool all Boolean circuits of polynomial size:

**Assumption 1.** *For any $c > 1$, there exists $k \in \mathbb{N}$ and $P : \{\pm 1\}^k \to \{\pm 1\}$ such that for $m$ sufficiently large, with probability $1 - o_m(1)$ over the randomness of $H$ in Definition 5, the function $G = G_{P,H}$ negl($m$)-fools all polynomial-size Boolean circuits relative to $U_{m^c}$ with seed $U_m$ for some negligible function negl $: \mathbb{N} \to [0,1]$.[4]*

## 2.3 DIVERSE DISTRIBUTIONS

Recall that the main result of this paper is to construct generators that look indistinguishable from natural target distributions $\mathcal{D}^*$ according to any poly-sized neural network, but which are far from $\mathcal{D}^*$ in Wasserstein. In this section we describe in greater detail the properties that these $\mathcal{D}^*$ satisfy.

**Definition 6.** *A distribution $\mu$ over $\mathbb{R}^d$ is $(N, \beta)$-diverse if for any discrete distribution $\nu$ on $\mathbb{R}^d$ supported on at most $N$ points, $W_1(\mu, \nu) \geq \beta$.*

The main fact we use about $(N, \beta)$-diverse distributions is that they cannot be approximated by pushforwards of $U_{\log_2 N}$ (see Lemma A.12 in the supplement for a formal statement).

Note that Definition 6 is a very mild assumption that simply requires that the distribution not be tightly concentrated around a few points. Distributions that satisfy Definition 6 are both practically relevant and highly expressive. For starters, any reasonable real-world image distribution will be diverse as it will not be concentrated around a few unique images. One can also show that distributions like uniform distributions over well-separated discrete point sets and over $[0,1]^d$ are diverse (Lemma A.13 and Lemma A.15 in supplement). Definition 6 also captures random expansive neural networks with leaky ReLU activations:

**Lemma 2.3** (Random expansive leaky ReLU networks). *For $k_0, \ldots, k_L \in \mathbb{N}$ satisfying $k_i \geq 1.1 k_{i-1}$ for all $i \in [L]$, let $\mathbf{W}_1 \in \mathbb{R}^{k_1 \times k_0}, \mathbf{W}_2 \in \mathbb{R}^{k_2 \times k_1}, \ldots, \mathbf{W}_L \in \mathbb{R}^{k_L \times k_{L-1}}$ be random weight matrices, where every entry of $\mathbf{W}_i$ is an independent draw from $\mathcal{N}(0, 1/k_i)$. For the function $F : \mathbb{R}^{k_0} \to \mathbb{R}^{k_L}$ given by $F(x) \triangleq \mathbf{W}_L \psi_\lambda (\mathbf{W}_{L-1} \psi_\lambda (\cdots \psi_\lambda (\mathbf{W}_1 x) \cdots))$, where $\psi_\lambda(z) = \psi_\lambda(z) = z/2 + (1/2 - \lambda)|z|$ is the leaky ReLU activation, $F(I_{k_0})$ is $(2^m, \beta)$-diverse for $m = (k_0/2) \log(k_0/2) - k_L/1.1 - 1$ and $\beta = \Theta(\lambda)^L$ with probability at least $1 - \exp(-\Omega(d))$.*

*For example, if $\lambda, L = \Theta(1)$, then $F(I_{k_0})$ is $(2^{\Omega(k_0 \log k_0)}, \Omega(1))$-diverse for $k_0$ sufficiently large.*

It is easy to see that the networks in Lemma 2.3 can be implemented as ReLU networks, so our main theorem applies to target distributions given by pushing the uniform distribution over the continuous cube through a random expansive leaky ReLU network of constant depth.

## 3 FOOLING RELU NETWORK DISCRIMINATORS DOES NOT SUFFICE

In this section we will show that even though a generative model looks indistinguishable from some target distribution $\mathcal{D}^*$ according to any ReLU network in $\mathcal{F}^*$, it can be quite far from $\mathcal{D}^*$ in Wasser-

---

[4]$g : \mathbb{N} \to \mathbb{R}_{\geq 0}$ is negligible if for every polynomial $p$, $g(n) < |1/p(n)|$ for all sufficiently large $n$.

stein. We begin by describing a simple version of this result over discrete domains in Section 3.1. In Section 3.2 we extend this to target distributions over continuous domains, but where the generator still takes in a discrete-valued seed. Finally, in Section 3.3 we give a simple reduction that extends these results to give generators that take in a continuous-valued random (Gaussian) seed, culminating in the following main result:

**Theorem 3.1.** *Let $\{H_m\}_m$ be a sequence of generators $H_m : \mathbb{R}^{r(m)} \rightarrow \mathbb{R}^{d(m)}$ for $r(m), d(m) \leq \mathrm{poly}(m)$ whose output coordinates are computable by networks in $\mathcal{C}_{L'(m),S'(m),r(m)}^{\tau'(m),\Lambda'(m)}$ for $\tau'(m), \Lambda'(m) \leq \mathrm{poly}(m)$. Suppose that $H_m(I_{r(m)})$ is $(2^m, \Omega(1))$-diverse.*

*Fix any $\epsilon : \mathbb{N} \rightarrow [0,1]$ satisfying $\epsilon(m) \geq \max(\mathsf{negl}(m), \exp(-O(m)))$. Under Assumption 1, there is a sequence of generators $G_m : \mathbb{R}^m \rightarrow \mathbb{R}^{d(m)}$ such that for all $m$ sufficiently large:*

1. *Every output coordinate of $G_m$ is computable by a network in $\mathcal{C}_{L,S,m}^{\tau,\Lambda}$ for*

$$\tau = \max(O(\log(\Lambda'(m) \cdot m \cdot d(m)/\epsilon(m))), \tau'(m), O(1)), \qquad \Lambda = O(\Lambda'(m)^2 \mathrm{poly}(m)/\epsilon(m)),$$

$$L = L'(m) + O(1), \qquad S = O(r(m)\log(1/\epsilon(m))) + 3m + S'(m).$$

2. *$W_{\mathcal{F}^*}(G_m(\gamma_m), H_m(I_{r(m)})) \leq \epsilon(m) \cdot \mathrm{poly}(m)$,*    3. *$W_1(G_m(\gamma_m), H_m(I_{r(m)})) \geq \Omega(1)$.*

Note that a natural choice of parameters for $H_m$ would be

$$\tau'(m) \leq O(\log m), \;\; \Lambda'(m), S'(m), 1/\epsilon(m) \leq \mathrm{poly}(m), \;\; L'(m) \leq O(1).$$

(In fact, the $\mathrm{poly}(m)$ factor in bullet point 2 simply comes from the Lipschitzness of $H_m$, so $\epsilon(m)$ only needs to scale inversely in this quantity for $W_{\mathcal{F}^*}$ to be small.) Altogether, we conclude that $G_m$'s output coordinates are computable by *constant-depth ReLU networks* with polynomial size and Lipschitzness and logarithmic bit complexity $\tau$.

## 3.1 STRETCHING BITS TO BITS

As a warmup, in this subsection we prove the following special case of Theorem 3.1 when the target distribution and seed distribution are discrete.

**Theorem 3.2.** *Under Assumption 1, for any constant $c > 1$, there is a sequence of generators $G_m : \mathbb{R}^m \rightarrow \mathbb{R}^{d(m)}$ for $d(m) \geq m^c$ such that for all $m$,*

1. *Every output coordinate of $G_m$ is computable by a network in $\mathcal{C}_{L,S,m}^{\tau,\Lambda}$ for $\tau, \Lambda, L, S = O_c(1)$.*

2. *$W_{\mathcal{F}^*}(G_m(U_m), U_{d(m)}) \leq \mathsf{negl}(m)$,*    3. *$W_1(G_m(U_m), U_{d(m)}) \geq \Omega(1)$.*

We emphasize that in this discrete setting, our quantitative guarantees are even stronger: *all parameters $\tau, \Lambda, L, S$ of the generator are constant*, and no polynomial-sized ReLU network can distinguish between $G_m(U_m)$ and $U_{d(m)}$ with even *non-negligible* advantage.

As discussed in the introduction, a basic but important building block in the proof of Theorem 3.2 is the connection between Goldreich's PRG and generative models computed by neural networks of constant depth/size/Lipschitzness. We begin by elaborating on this connection and showing that any predicate $\{\pm 1\}^k \rightarrow \{\pm 1\}$ can be implemented as a network in $\mathcal{C}_{L,S,d}^{\tau,\Lambda}$ where $\tau, \Lambda, L, S = O_k(1)$. As a consequence of known constructions for implementing Boolean functions with ReLU networks (see e.g. Lemma A.2 of (Chen et al., 2020b)), we can show:

**Corollary 3.3.** *For any $k \in \mathbb{N}$ and any $H$ and $P$ as in Definition 5, the coordinates of the output of $G_{P,H}$ are computed by networks in $\mathcal{C}_{L,S,m}^{\tau,\Lambda}$ for $\tau = O(k), \Lambda = \exp(O(k)), L = k, S = O(2^k k)$.*

Before we use this to prove Theorem 3.2, we need an extra technical ingredient to formalize the fact that a discriminator given by a ReLU network of polynomially bounded complexity yields a discriminator computable by a polynomial-sized Boolean circuit. The idea is that if $W_{\mathcal{F}^*}$ is large so that there exists some ReLU network discriminator, then because the input to the discriminator is sufficiently well concentrated, some affine threshold of the ReLU network can distinguish between the two distributions:

**Lemma 3.4.** *Given independent $X$ and $Y$ such that $\mathbb{E}[Y] - \mathbb{E}[X] = \alpha$ and for which $X - \mathbb{E}[X]$ and $Y - \mathbb{E}[Y]$ are $\sigma^2$-sub-Gaussian, there exists a threshold $t \in [\mathbb{E}[X] - O(\sigma\sqrt{\log(\sigma/|\alpha|)}), \mathbb{E}[Y] + O(\sigma\sqrt{\log(\sigma/|\alpha|)})]$ for which $|\mathbb{P}[X > t] - \mathbb{P}[Y > t]| \geq \min\left(1/2, \widetilde{\Omega}(|\alpha|/\sigma)\right)$.*

We can now sketch the proof of Theorem 3.2, deferring a formal argument to the supplement.

*Proof sketch of Theorem 3.2.* The parameter $m$ will be clear from context in the following discussion, so for convenience we will refer to $d(m)$ and $G_m$ as $d$ and $G$. Let $k, P, G, \mathsf{negl}(\cdot)$ be such that the outcome of Assumption 1 holds. By Corollary 3.3, every output coordinate of $G$ is computable by a network in $\mathcal{C}_{L,S,m}^{\tau,\Lambda}$ for $\tau = O(k), \Lambda = \exp(O(k)), L = k, S = O(2^k k)$.

The fact that $W_1(G(U_m), U_d) > 1/3$ follows from the fact that $G(U_m)$ is uniform over $2^m$ points on the hypercube (with multiplicity), and $U_d$ is $(2^m, 1/3)$-diverse.

It remains to check that $G$ fools $\mathcal{F}^*$ relative to $U_d$. Suppose to the contrary that there exists some $f : \mathbb{R}^d \to \mathbb{R}$ in $\mathcal{F}^*$ for which $|\mathbb{E}[f(G(U_m))] - \mathbb{E}[f(U_d)]| > d^{-a}$ for some constant $a > 0$. Note that for any threshold $t \in \mathbb{R}_\tau$, there is a Turing machine $\mathcal{M}_\tau : \{\pm 1\}^d \to \{\pm 1\}$ that computes $y \mapsto \mathrm{sgn}(f(y) - t)$ using $\tau$ bits of advice. By applying Lemma 3.4 to random variables $X = f(G(U_m))$ and $Y = f(U_d)$, we get that there exists a threshold $t \in \mathbb{R}_{\mathrm{poly}(d)}$ for which $|\mathbb{E}[\mathcal{M}_\tau(G(U_m))] - \mathbb{E}[\mathcal{M}_\tau(U_d)]| > 1/\mathrm{poly}(d)$, contradicting Assumption 1. $\square$

## 3.2 FROM BINARY OUTPUTS TO CONTINUOUS OUTPUTS

In this section we show how to extend Theorem 3.2 to the setting where the target distribution $\mathcal{D}^*$ is a pushforward of the uniform distribution on $[0, 1]^r$. At a high level, the idea will be to *post-process* the output of the generator constructed in Theorem 3.2. Roughly speaking, we take weighted averages of clusters of output coordinates from the generator in Theorem 3.2 and pass these averages through the pushforward map defining $\mathcal{D}^*$. Formally, we show:

**Theorem 3.5.** *Let $\{H_m\}_m$ be a sequence of generators $H_m : \mathbb{R}^{r(m)} \to \mathbb{R}^{d(m)}$ for $r(m), d(m) \leq \mathrm{poly}(m)$ whose output coordinates are computable by networks in $\mathcal{C}_{L'(m),S'(m),r(m)}^{\tau'(m),\Lambda'(m)}$ for $\tau'(m), \Lambda'(m) \leq \mathrm{poly}(m)$. Suppose that $H_m(I_{r(m)})$ is $(2^m, \Omega(1))$-diverse.*

*Fix any $\epsilon : \mathbb{N} \to [0, 1]$ satisfying $\epsilon(m) \geq \max(\mathsf{negl}(m), \exp(-O(m)))$. Under Assumption 1, there is a sequence of generators $G_m : \mathbb{R}^m \to \mathbb{R}^{d(m)}$ such that for all $m$ sufficiently large:*

*1. Every output coordinate of $G_m$ is computable by a network in $\mathcal{C}_{L,S,m}^{\tau,\Lambda}$ for*

$$\tau = \max(O(\log(1/\epsilon(m))), \tau'(m), O(1)), \qquad \Lambda = O(\Lambda'(m)) \cdot \mathrm{poly}(m),$$
$$L = L'(m) + O(1), \qquad S = O(r(m) \cdot \log(1/\epsilon(m))) + S'(m).$$

*2. $W_{\mathcal{F}^*}(G_m(U_m), H_m(I_{r(m)})) \leq \epsilon(m) \cdot \mathrm{poly}(m)$,     3. $W_1(G_m(U_m), H_m(I_{r(m)})) \geq \Omega(1)$.*

Theorem 3.5 retains many of the nice properties of Theorem 3.2, e.g. it can tolerate distinguishing advantage $\epsilon(m)$ which is negligible, at the mild cost of an extra logarithmic dependence on $1/\epsilon(m)$ in the bit complexity $\tau$. And as with Theorem 3.1, if the networks $H_m$ are of constant depth, the resulting generators $G_m$ are also of constant depth.

To prove Theorem 3.5, we begin by showing that for any pair of distributions which are close under the $W_{\mathcal{F}}$ metric for some family of neural networks $\mathcal{F}$, their pushforwards under a simple generative model will still be close under the $W_{\mathcal{F}'}$ metric for some slightly weaker family of networks $\mathcal{F}'$.

**Lemma 3.6.** *Fix parameters $\Lambda, \Lambda' > 1$. Let $\mathcal{F} = \mathcal{C}_{L,S,s}^{\tau,\Lambda}$. If $W_{\mathcal{F}}(p, q) \leq \epsilon$ for some distributions $p, q$ on $\mathbb{R}^s$, then for any $J : \mathbb{R}^s \to \mathbb{R}^r$ each of whose output coordinates is computed by a function in $\mathcal{C}_{L',S',s}^{\tau,\Lambda'}$ for some $L' < L$ and $S' \leq \frac{S-r-1}{r}$, we have $W_{\mathcal{F}'}(J(p), J(q)) \leq 2\epsilon\Lambda'\sqrt{r}$ for $\mathcal{F}' = \mathcal{C}_{L-L',S'',r}^{\tau',\Lambda}$ where $\tau' = \tau - \lceil\log_2 \Lambda\sqrt{r}\rceil$ and $S'' = S - r(S' + 1)$.*

Now recall that in Theorem 3.2, we exhibited a GAN which is close in $W_{\mathcal{F}^*}$ to $U_s$. Using Lemma 3.6, we can show that a certain simple pushforward of this GAN will be close in $W_{\mathcal{F}^*}$ to *the uniform distribution over $[0, 1]^r$* for $r$ slightly smaller than $s$. The starting point is the following:

**Fact 3.7.** *For any $0 < \epsilon < 1$ and $n \geq \log_2(1/\epsilon)$, let $h : \mathbb{R}^n \to \mathbb{R}$ be given by $h(x) = \langle w, x + \mathbf{1} \rangle$ for $w = \left(1/4, 1/8, \ldots, (1/2)^{n+1}\right)$ and $\mathbf{1}$ the all-1's vector. Then $W_1(h(U_n), I_1) \leq \epsilon$.*

By leveraging Lemma 3.6 and Fact 3.7, we can get an approximation to the uniform distribution over $[0, 1]^r$ out of the uniform distribution over $\{\pm 1\}^s$:

**Lemma 3.8.** *Suppose $\epsilon > 0$ satisfies $\log(1/\epsilon) \leq \text{poly}(s)$. If a distribution $\widetilde{\mathcal{D}}$ over $\mathbb{R}^s$ satisfies $W_{\mathcal{F}^*}(\widetilde{\mathcal{D}}, U_s) \leq \epsilon$, then for $r \triangleq s/\lceil \log(1/\epsilon) \rceil$,[5] there is a function $J : \mathbb{R}^s \to \mathbb{R}^r$ each of whose output coordinates is computed by a function in $\mathcal{C}_{1,0,s}^{O(\log(1/\epsilon)),O(1)}$ such that $W_{\mathcal{F}^*}(J(\widetilde{\mathcal{D}}), I_r) \leq \epsilon \cdot \text{poly}(r)$.*

By further combining Lemma 3.6 and Lemma 3.8, we can thus extend the latter from the uniform distribution on $[0, 1]^r$ to simple pushforwards thereof.

**Lemma 3.9.** *Under the hypotheses of Lemma 3.8, for any $d \leq \text{poly}(s)$ and any function $H : \mathbb{R}^r \to \mathbb{R}^d$ each of whose output coordinates is computed by a function in $\mathcal{C}_{L',S',r}^{\tau',\Lambda'}$ for $\tau' \leq \text{poly}(s)$, there is a function $J' : \mathbb{R}^s \to \mathbb{R}^d$ each of whose output coordinates is computed by a function in $\mathcal{C}_{L'+1,r+S',s}^{\max(O(\log(1/\epsilon)),\tau'),O(\Lambda'\sqrt{d})}$ such that $W_{\mathcal{F}^*}(J'(\widetilde{\mathcal{D}}), H(I_r)) \leq \epsilon\Lambda' \cdot \text{poly}(s)$.*

So for any pushforward $H(I_r)$ of the uniform distribution on $[0, 1]^r$, Lemma 3.9 lets us take the GAN given by Theorem 3.2 and slightly post-process its output so that it is close in $W_{\mathcal{F}^*}$ to $H(I_r)$. We can now sketch Theorem 3.5's proof, deferring the full argument to the supplement.

*Proof of Theorem 3.5.* Condition 3 follows because $G(U_m)$ is a uniform distribution on $2^m$ points (w/ multiplicity) and $H(I_r)$ is $(2^m, \Omega(1))$-diverse. Next, let $s = r \cdot \lceil \log(1/\epsilon) \rceil$. As we are assuming $\epsilon \geq \exp(-O(m))$, $s \leq r \cdot m = m^c$ for some constant $c > 1$. Take $G'$ to be the generator $G : \mathbb{R}^m \to \mathbb{R}^s$ constructed in Theorem 3.2. By applying Lemma 3.9 to $\widetilde{\mathcal{D}} = G'(U_m)$, we get a function $J' : \mathbb{R}^s \to \mathbb{R}^d$ each of whose output coordinates is computed by a function in $\mathcal{C}_{L'+1,r+S',s}^{\max(O(\log(1/\epsilon)),\tau'),O(\Lambda'\sqrt{d})}$ such that $W_{F_d}(J'(G'(U_m)), H(I_r)) \leq \epsilon\Lambda' \cdot \text{poly}(m) \leq \epsilon \cdot \text{poly}(m)$, so we get condition 2 of the theorem for $G \triangleq J' \circ G'$. Finally, by Lemma 2.1, every output coordinate of $G$ can be realized by a network in $\mathcal{C}_{L,S,m}^{\tau,\Lambda}$ for $\tau = \max(O(\log(1/\epsilon)),\tau',O(1))$, $\Lambda = O(\Lambda'\sqrt{ds}) = O(\Lambda') \cdot \text{poly}(m)$, $L = L' + O(1)$, and $S = O(s) + r + S' = O(s) + S'$ (where we used the fact that $r = s/\lceil \log(1/\epsilon) \rceil < s$). This establishes condition 1 of the theorem. □

### 3.3 FROM BINARY INPUTS TO CONTINUOUS INPUTS

In Theorem 3.5, we have shown how to go from $U_m$ to any simple pushforward $H_m(I_{r(m)})$ of the uniform distribution over $[0, 1]^{r(m)}$. Here we complete the proof of our main result, Theorem 3.1, by giving a simple reduction showing how to use *Gaussian seed* $\gamma_m$ instead of $U_m$. At a high level, the idea will be to *pre-process* the inputs to the generator constructed in Theorem 3.5 by appending appropriate activations at the input layer. We will need the following elementary construction.

**Lemma 3.10.** *For any $\xi$ for which $1/\xi \in \mathbb{R}_\tau$, the piecewise linear function $h_\xi$ which outputs $-1$ for inputs $x \leq -\xi$, $1$ for $x \geq \xi$, and $x/\xi$ otherwise, lies in $\mathcal{C}_{2,2,1}^{\tau,1/\xi}$.*

The function $h_\xi$ will let us approximately convert from $\gamma_m$ to $U_m$. Specifically, the following says that if we want to approximate the output distribution of the generator in Theorem 3.5 using Gaussians instead of bits as seed, it suffices to attach entrywise applications of $h_\xi$ at the input layer:

**Lemma 3.11.** *Let $G_0 : \mathbb{R}^m \to \mathbb{R}^d$ have output coordinates computable by networks in $\mathcal{C}_{L'',S'',m''}^{\tau'',\Lambda''}$. For any $\epsilon > 0$, let $\xi' \triangleq \epsilon/\left(\Lambda''\sqrt{(2/\pi)m^3 d}\right)$ and let $\xi$ bethe multiplicative inverse of $\lceil 1/\xi' \rceil$.*

*The function $G : \mathbb{R}^m \to \mathbb{R}^d$ given by $G(x) = G_0(h_\xi(x))$, where $h_\xi(x)$ denotes entrywise application of $h_\xi$ defined in Lemma 3.10, satisfies that 1) each of the output coordinates of $G$ is computable by a network in $\mathcal{C}_{L,S,m}^{\tau,\Lambda}$ for the parameters $\tau = \max(\tau'', O(\log(\Lambda'' md/\epsilon)))$, $\Lambda = O(\Lambda''^2 m^2 \sqrt{d}/\epsilon)$, $L = L'' + 2$, and $S = 3m + S''$, and 2) $W_1(G_0(U_m), G(\gamma_m)) \leq \epsilon$.*

---

[5] We will assume for simplicity that this is an integer, though it is not hard to handle the case where $\lceil \log(1/\epsilon) \rceil$ does not divide $s$.

*Remark* 3.12. Note the only fact we use about $\gamma_m$ in the proof of Lemma 3.11 is that outside of an event with probability $O(m\xi)$, $h_\xi(\gamma_m)$ is uniform over $\{\pm 1\}^m$. In particular, the same would hold for any product measure each of whose coordinates is symmetric and anticoncentrated around zero. For instance, up to a constant factor in $\xi'$, Lemma 3.11 also holds with $\gamma_m$ replaced by $I_m$.

*Proof of Theorem 3.1.* Substitute the generator constructed in Theorem 3.5, call it $G_0$, into Lemma 3.11; let $G$ be the generator resulting from the lemma. Recall from Theorem 3.5 that each output coordinate of $G_0$ lies in $\mathcal{C}_{L'',S'',m}^{\tau'',\Lambda''}$ for $\tau'' = \max(O(\log(1/\epsilon)),\tau',O(1))$, $\Lambda'' = O(\Lambda'\mathrm{poly}(m))$, $L'' = L' + O(1)$, $S'' = O(r\log(1/\epsilon)) + S'$. So by Lemma 3.11, every output coordinate of $G$ is computable by a network in $\mathcal{C}_{L,S,m}^{\tau,\Lambda}$ for $\tau = \max(\tau'',O(\log(\Lambda'md/\epsilon))) = \max(O(\log(\Lambda md/\epsilon)),\tau',O(1))$, $\Lambda = O(\Lambda''^2m^2\sqrt{d}/\epsilon) = O(\Lambda'^2\mathrm{poly}(m)/\epsilon)$, $L = L'' + 2 = L' + O(1)$, and $S = 3m + S'' = O(r\log(1/\epsilon)) + 3m + S''$. $\qquad\square$

## 4 EXPERIMENTAL RESULTS

To empirically demonstrate the existence of a constant depth generator that can fool polynomially-bounded discriminators, we evaluated the generator $G$ given by Goldreich's PRG (Goldreich, 2011) with input dimension $m = 50$, output dimension $d = 200$, and predicate $P : \{\pm 1\}^5 \to \{\pm 1\}$ given by the popular TSA predicate, namely $P(x_1,\ldots,x_5) = x_1 \cdot x_2 \cdot x_3 \cdot (x_4 \wedge x_5)$. This is the smallest predicate under which Goldreich's candidate construction is believed to be secure.

The target distribution $\mathcal{D}^*$ is the uniform distribution $U_{200}$ over $\{\pm 1\}^{200}$. As we prove in Lemma A.13, $U_d$ is sufficiently diverse that $W_1(G(U_m), U_d) \geq \Omega(1)$. We trained four different discriminators given respectively by $1, 2, 3, 4$ hidden-layer ReLU networks, where each hidden layer is fully connected with dimensions $200 \times 200$, to discriminate the output of the generator $G(U_m)$ from the target distribution $U_d$. We used the Adam optimizer with step size $0.001$ over the DCGAN training objective, with batch-size $128$. As we can see in Figure 1, the test loss $\mathbb{E}[-\log(D(X))] + \mathbb{E}[-\log(1 - D(G(z)))] - 2\log(2)$ stays consistently above zero, indicating that the discriminator can not discriminate the true distribution from the generator output, even though the Wasserstein distance between these two distributions is provably large.

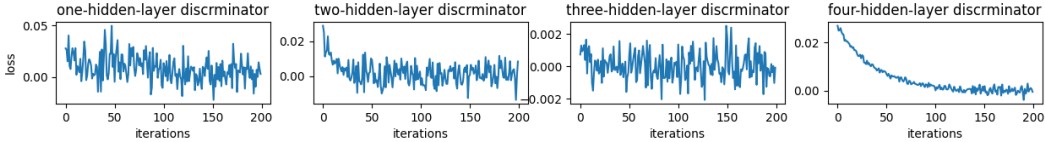

Figure 1: Test loss ($\mathbb{E}[-\log(D(X))] + \mathbb{E}[-\log(1 - D(G(z)))] - 2\log(2)$) over course of training. Discriminators cannot distinguish generator output from true distribution, though Wasserstein provably large.

## 5 CONCLUSIONS

In light of the obstructions presented in this paper, what are natural next steps for the theory of GANs? Here, we offer a couple of thoughts and possible future directions.

One limitation of our lower bound is that it holds for a specific generator. Of course, it is quite unlikely that we will ever encounter such a generator through natural GAN training. One way to circumvent our lower bound is to argue that the training dynamics of the generator may have some regularization effect which allows us to avoid these troublesome generators, and which allows GANs to learn distributions in polynomial time.

Another orthogonal perspective is that our results suggest that perhaps statistical learning is too strong of a goal. If our GAN is indeed indistinguishable from the target distribution to all polynomial time algorithms, then not only should the output of the GAN be sufficient for humans, but it should also be sufficient for all downstream applications, which presumably run in polynomial time. This raises the intriguing possibility that the correct metric for measuring closeness between distances in the context of GANs should inherently involve some computational component (e.g. in the sense of Dwork et al. (2021)) as opposed to the purely statistical metrics generally considered in the literature.

**Acknowledgments**   The authors would like to thank Boaz Barak, Adam Klivans, and Alex Lombardi for enlightening discussions about local PRGs. SC was supported by NSF Award 2103300. This work was done while the first, second, and fourth authors were visiting the Simons Institute for the Theory of Computing.

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

## A    ADDITIONAL TECHNICAL PRELIMINARIES

**More Notation and Miscellaneous Tools**    Let $\mathbf{1}_n$ denote the all-ones vector in $n$ dimensions; when $n$ is clear from context, we denote this by $\mathbf{1}$.

Given a vector $v \in \mathbb{R}^d$, we let $\|v\|$ denote its Euclidean norm. Given $r > 0$, let $B(v, r) \subset \mathbb{R}^d$ denote the Euclidean ball of radius $r$ with center $v$. Given a matrix $\mathbf{W}$, we let $\|\mathbf{W}\|$ denote its operator norm. Let $\sigma_{\min}(\mathbf{W})$ denote its minimum singular value.

Given a distribution $p$, let $p^{\otimes n}$ denote the product measure given by drawing $n$ independent samples from $p$.

Define the function $\mathrm{sgn}(x) \triangleq \begin{cases} 1 & x \geq 0 \\ -1 & x < 0 \end{cases}$.

Let $\phi : \mathbb{R} \to \mathbb{R}$ denote the ReLU activation $\phi(z) \triangleq \max(0, z)$. Let $\psi_\lambda : \mathbb{R} \to \mathbb{R}$ denote the leaky ReLU activation $\psi_\lambda(z) = z/2 + (1/2 - \lambda)|z|$. Note that

$$\psi_\lambda(z) = (1 - \lambda)\phi(z) - \lambda\phi(-z).$$

We will need the following well-known result:

**Theorem A.1** (Kirszbraun extension). *Given an arbitrary subset $S \subset \mathbb{R}^d$ and $f : S \to \mathbb{R}$ which is $L$-Lipschitz, there exists an $L$-Lipschitz extension $\widetilde{f} : \mathbb{R}^d \to \mathbb{R}$ for which $\widetilde{f}(y) = f(y)$ for all $y \in S$.*

We will need the following basic fact about composing Lipschitz functions:

**Fact A.2.** *If $g_1, \ldots, g_r : \mathbb{R}^d \to \mathbb{R}$ are $\Lambda$-Lipschitz and $h : \mathbb{R}^r \to \mathbb{R}$ is $\Lambda'$-Lipschitz, then the function*

$$x \mapsto h(g_1(x), \ldots, g_r(x))$$

*is $\Lambda\Lambda'\sqrt{r}$-Lipschitz.*

*Proof.* For any $x, x'$, we have $|g_i(x) - g_i(x')| \leq \Lambda\|x - x'\|$, so $\left(\sum_{i=1}^r (g_i(x) - g_i(x'))^2\right)^{1/2} \leq \Lambda\sqrt{r}\|x - x'\|$. This implies that $|h(g_1(x), \ldots, g_r(x)) - h(g_1(x'), \ldots, g_r(x'))| \leq \Lambda\Lambda'\sqrt{r}\|x - x'\|$ as desired. $\qquad\square$

**Fact A.3.** *The volume of a $d$-dimensional Euclidean ball of radius 1 is at most $(18/d)^{d/2}$.*

*Proof.* It is a standard fact that the volume of the ball can be expressed as $2^d \cdot \frac{(\pi/2)^{\lfloor d/2 \rfloor}}{d!!}$. If $d$ is even, then $d!! = 2^{d/2} \cdot (d/2)! \geq 2^{d/2} \cdot e\left(\frac{d}{2e}\right)^{d/2} \geq (d/e)^{d/2}$. If $d$ is odd, then $d!! = \frac{d!}{\lfloor d/2 \rfloor! \cdot 2^{\lfloor d/2 \rfloor}} \geq \frac{e(d/e)^d}{e(d/2e)^{d/2} \cdot 2^{d/2}} = (d/e)^{d/2}$. We conclude that the volume is at most $(2\pi e/d)^{d/2} \leq (18/d)^{d/2}$. $\qquad\square$

### A.1    CONCENTRATION OF MEASURE

We will use the following consequence of McDiarmid's inequality:

**Lemma A.4** (McDiarmid's Inequality). *Suppose $F : \{\pm 1\}^n \to \{\pm 1\}$ is such that for any $x, x' \in \{\pm 1\}^n$ differing on exactly one coordinate, $|F(x) - F(x')| \leq c$. Then*

$$\underset{x \sim \{\pm 1\}^n}{\mathbb{P}}[|f(x) - \mathbb{E}[f(x)]| > s] \leq \exp\left(-\frac{2s^2}{nc^2}\right).$$

**Corollary A.5.** *Given $F : \{\pm 1\}^n \rightarrow \{\pm 1\}$ which is $\Lambda$-Lipschitz, define the random variable $X \triangleq F(U_n)$. Then $X - \mathbb{E}[X]$ is $\Lambda\sqrt{2n}$-sub-Gaussian.*

*Proof.* Because $F$ is Lipschitz, it satisfies the hypothesis of Lemma A.4 with $c = \Lambda$, so the corollary follows by the definition of sub-Gaussianity. □

**Theorem A.6** (Theorem 1.1, (Rudelson & Vershynin, 2009))**.** *For $n, d \in \mathbb{N}$ with $n \geq d$, let $\mathbf{W} \in \mathbb{R}^{n \times d}$ be a random matrix whose entries are independent draws from $\mathcal{N}(0,1)$. Then for every $\epsilon > 0$,*

$$\mathbb{P}\Big[\sigma_{\min}(\mathbf{W}) \leq \epsilon(\sqrt{n} - \sqrt{d-1})\Big] \leq (C\epsilon)^{n-d+1} + e^{-cn}$$

*for absolute constants $C, c > 0$.*

### A.2 Boolean Circuits

In the context of pseudorandom generators, the set of all polynomial-sized Boolean circuits is the canonical family of discriminator functions to consider when formalizing what it means for a generator to fool all polynomial-time algorithms.

Here we review some basics about Boolean circuits; for a more thorough introduction to these concepts, we refer the reader to any of the standard textbooks on complexity theory, e.g. (Arora & Barak, 2009; Sipser, 1996).

**Definition 7** (Boolean circuits)**.** *Fix a set $G$ of logical gates, e.g. $\wedge, \vee, \neg$. A Boolean circuit $C$ is a Boolean function $\{\pm 1\}^n \rightarrow \{\pm 1\}$ given by a directed acyclic graph with $n$ input nodes with in-degree zero and an output node with out-degree zero, where each node that isn't an input node is labeled by some logical gate in $G$. Unless otherwise specified, we will take $G$ to be $\{\wedge, \vee, \neg\}$.*

*The* size *$S$ of the circuit is the number of nodes in the graph, and the* depth *$D$ is given by the length of the longest directed path in the graph. The value of $C$ on input $x \in \{\pm 1\}^n$ is defined in an inductive fashion: the value at a node $v$ in the graph is defined to be the evaluation of the gate at $v$ on the in-neighbors of $v$ (as the graph is acyclic, this is well-defined), and the value of $C$ on $x$ is then the value of the output node.*

*We will occasionally also be interested in the number $W$ of* wires *in the circuit, i.e. the number of edges in the graph. Note that trivially*

$$S \leq W + 1. \tag{2}$$

**Definition 8** (P/poly)**.** *Given $T : \mathbb{N} \rightarrow \mathbb{N}$, let $\mathsf{SIZE}(T(n))$ denote the family of sequences of Boolean functions $\{f_n : \{\pm 1\}^n \rightarrow \{\pm 1\}\}$ for which there exist Boolean circuits $\{C_n\}$ with sizes $\{S_n\}$ that compute $\{f_n\}$ and such that $S_n \leq T(n)$.*

*Let $\mathsf{P/poly} \triangleq \bigcup_{c>1} \mathsf{SIZE}(n^c)$. We refer to (sequences of) functions in $\mathsf{P/poly}$ as functions computable by polynomial-sized circuits.*

The following standard fact about bounded-depth Boolean circuits will make it convenient to translate between them and neural networks.

**Lemma A.7** (See Theorem 1.1 in Section 12.1 of (Wegener, 1987))**.** *For any Boolean circuit $C$ of size $S$ and depth $D$ with gate set $G$, there is another circuit $C'$ of size $D \cdot S$ and depth $D$ with gate set $G$ which computes the same function as $C$ but with the additional property that for any gate in $C'$, all paths from an input to the gate are of the same length.*

The upshot of Lemma A.7 is that for any length $\ell$, we can think of the gates of $C'$ at distance $\ell$ from the inputs as comprising a "layer" in the circuit.

A less combinatorial way of formulating the complexity class captured by polynomial-sized circuits is in terms of Turing machines with *advice strings*.

**Fact A.8** (See e.g. Theorem 6.11 in (Arora & Barak, 2009))**.** *A sequence of Boolean functions $\{f_n : \{\pm 1\}^n \rightarrow \{\pm 1\}\}$ is in $\mathsf{P/poly}$ if and only if there exists a sequence of* advice strings $\{\alpha_n\}$, *where $\alpha_n \in \{\pm 1\}^n$ for $a_n \leq \mathrm{poly}(n)$, and a Turing machine $M$ which runs for at most $\mathrm{poly}(n)$ steps and, for any $n \in \mathbb{N}$, takes as input any $x \in \{\pm 1\}^n$ and the advice string $\alpha_n$ and outputs $M(x, \alpha_n) = f_n(x)$.*

This fact will be useful for translating discriminators computed by neural networks into discriminators given by polynomial-sized Boolean circuits.

## A.3 MORE ON GANS AND PRGS

In this section we fill in some additional details regarding the contents of Section 2.1. We begin with a simple remark about Definition 1.

*Remark* A.9. In Definition 1, if $L = 1$, then $S = 0$ and the definition specializes to linear functions. That is, $\mathcal{C}_{1,0,d}^{\tau,\Lambda}$ is simply the class of affine linear functions $F(x) = \langle w, x \rangle + b$ for $w \in \mathbb{R}_\tau^d$ and $b \in \mathbb{R}_\tau$ satisfying $\|w\| \leq \Lambda$.

Next, we fill in the proof of Lemma 2.1, which we restate below for the reader's convenience.

**Lemma A.10.** *Let $J : \mathbb{R}^s \to \mathbb{R}^r$ be a function each of whose output coordinates is computed by some network in $\mathcal{C}_{L_1,S_1,s}^{\tau_1,\Lambda_1}$, and let $f \in \mathcal{C}_{L_2,S_2,r}^{\tau_2,\Lambda_2}$. Then $f \circ J \in \mathcal{C}_{L,S,s}^{\tau,\Lambda}$ for $\tau = \max(\tau_1, \tau_2)$, $\Lambda = \Lambda_1 \Lambda_2 \sqrt{r}$, $L = L_1 + L_2$, and $S = (S_1 + 1)r + S_2$. Furthermore, for the network in $\mathcal{C}_{L,S,s}^{\tau,\Lambda}$ realizing $f \circ J$, the bias and weight vector entries in the output layer lie in $\mathbb{R}_{\tau_2}$.*

*Proof.* Suppose that the $i$-th output coordinate of $J$ is computed by a neural network with weight matrices $\mathbf{W}_1^{(i)} \in \mathbb{R}^{k_1^{(i)} \times s}, \ldots, \mathbf{W}_{L_1}^{(i)} \in \mathbb{R}^{1 \times k_{L_1-1}^{(i)}}$ and biases $b_1^{(i)} \in \mathbb{R}^{k_1^{(i)}}, \ldots, b_{L_1}^{(i)} \in \mathbb{R}$.

Define the $(\sum_{i=1}^r k_1^{(i)}) \times s$ weight matrix $\mathbf{W}_1$ by vertically concatenating the weight matrices $\mathbf{W}_1^{(1)}, \ldots, \mathbf{W}_1^{(r)}$. For every $1 < j < L_1$ define the $(\sum_{i=1}^r k_j^{(i)}) \times (\sum_{i=1}^r k_{j-1}^{(i)})$ weight matrix $\mathbf{W}_j$ by diagonally concatenating the weight matrices $\mathbf{W}_j^{(1)}, \ldots, \mathbf{W}_j^{(r)}$. Similarly, define the $r \times (\sum_{i=1}^r k_{L_1}^{(i)})$ matrix $\mathbf{W}_{L_1}$ by diagonally concatening the column vectors $\mathbf{W}_{L_1}^{(1)}, \ldots, \mathbf{W}_{L_1}^{(r)}$. For the bias vectors in these layers, for every $1 \leq j \leq L_1$ define $b_j$ to be the vector given by concatenating $b_j^{(1)}, \ldots, b_j^{(r)}$.

Now suppose that $f$ is computed by a neural network with weight matrices $\mathbf{W}_{L_1+1} \in \mathbb{R}^{k_{L_1+1} \times r}$, $\ldots, \mathbf{W}_{L_1+L_2} \in \mathbb{R}^{1 \times k_{L_1+L_2-1}}$ and biases $b_{L_1+1} \in \mathbb{R}^{k_{L_1+1}}, \ldots, b_{L_1+L_2} \in \mathbb{R}$. Then by design, for any $y \in \mathbb{R}^s$ we have

$$f(J(y)) = \mathbf{W}_{L_1+L_2} \phi(\mathbf{W}_{L_1+L_2-1} \phi(\cdots \phi(\mathbf{W}_1 y + b_1) \cdots) + b_{L_1+L_2-1}) + b_{L_1+L_2}.$$

This network has depth $L_1 + L_2$ and size

$$\left( \sum_{j=1}^{L_1-1} \sum_{i=1}^r k_j^{(i)} \right) + r + \sum_{j=L_1+1}^{L_1+L_2} k_j = r \cdot S_1 + r + S_2 = S.$$

The bit complexity of the entries of the weight matrices and biases are obviously bounded by $\max(\tau_1, \tau_2)$, and the Lipschitzness of the network is bounded by $\Lambda_1 \Lambda_2 \sqrt{r}$ by Fact A.2. $\square$

Finally, we will also use the following standard tensorization property of Wasserstein distance later:

**Fact A.11** (See e.g. Lemma 3 in (Mariucci & Reiß, 2018)). *If $p, q$ satisfy $W_1(p, q) \leq \epsilon$, then $W_1(p^{\otimes n}, q^{\otimes n}) \leq \epsilon \sqrt{n}$.*

## A.4 DIVERSE DISTRIBUTIONS: MISSING PROOFS

Here we fill in some missing details from Section 2.3. We first show that diverse distributions cannot be approximated by pushforwards of $U_m$ if $m$ is insufficiently large. This follows immediately from the definition of diversity:

**Lemma A.12.** *For any $0 < \beta < 1$, if $\mathcal{D}^*$ is a $(2^m, \beta)$-diverse distribution over $\mathbb{R}^d$, then for any function $G : \{\pm 1\}^m \to \mathbb{R}^d$, $W_1(G(U_m), \mathcal{D}^*) \geq \beta$.*

*Proof.* $G(U_m)$ is a uniform distribution on $2^m$ points, with multiplicity if there are multiple points in $\{\pm 1\}^m$ that map to the same point in $\mathbb{R}^d$ under $G$, so the claim follows by definition of diversity. $\square$

Below we give some simple examples of diverse distributions.

**Lemma A.13** (Discrete, well-separated distributions). *For any $\alpha > 0$ and any $N, N' \in \mathbb{N}$ satisfying $N \leq N'$. Let $\Omega \subseteq \mathbb{R}^d$ be a set of points such that for any $z, z' \in \Omega$, $\|z - z'\| \geq \alpha$. Then the uniform distribution $\mu$ on any $N'$ points from $\Omega$ is $(N, \beta)$-diverse for $\beta = \alpha(1 - N/N')$.*

*Proof.* Take any discrete distribution $\nu$ supported on at most $N$ points $y_1, \ldots, y_N$ in $\mathbb{R}^d$. Consider the function $f : \mathbb{R}^d \to \mathbb{R}$: for any $y$ in the support of $\nu$, let $f(y) = 0$, and for any $y$ not in the support of $\nu$, let $f(y) = 1$. As a function from $\Omega$ to $\mathbb{R}$, where $\Omega$ inherits the Euclidean metric, $f$ is clearly $1/\alpha$-Lipschitz over $\Omega$. By Theorem A.1, there exists a $1/\alpha$-Lipschitz extension $\widetilde{f} : \mathbb{R}^d \to \mathbb{R}$ of $f$, and we have

$$|\mathbb{E}[f(\mu)] - \mathbb{E}[f(\nu)]| = |\mathbb{E}[f(\mu)]| \geq 1 - N/N',$$

so $W_1(\mu, \nu) \geq 1 - N/N'$ as desired. $\square$

We now turn to examples of continuous distributions which are diverse. We first observe that a distribution is $(N, \beta)$-diverse if it satisfies certain small-ball probability bounds.

**Definition 9.** *For a distribution $\mathcal{D}$ over $\mathbb{R}^d$, define the Lévy concentration function $Q_{\mathcal{D}}(r) \triangleq \sup_{x' \in \mathbb{R}^d} \mathbb{P}_{x \sim \mathcal{D}}[\|x - x'\| \leq r]$.*

**Lemma A.14.** *If a distribution $\mathcal{D}$ over $\mathbb{R}^d$ satisfies $Q_{\mathcal{D}}(r) \leq \alpha$, then $\mathcal{D}$ is $(N, r(1 - N\alpha))$-diverse.*

*Proof.* Take any $N$ points $z_1, \ldots, z_N \in \mathbb{R}^d$. By the bound on $Q_{\mathcal{D}}(r)$, the union $S$ of the balls of radius $r$ around these points has Lebesgue measure at most $N\alpha$. Define the function $f : \{z_1, \ldots, z_N\} \cup (\mathbb{R}^d \backslash S) \to \{0, 1\}$ to be zero on $\{z_1, \ldots, z_N\}$ and one on $\mathbb{R}^d \backslash S$. This function is $1/r$-Lipschitz on its domain, so by Theorem A.1 there is an extension $f' : \mathbb{R}^d \to \mathbb{R}$ of $f$ which remains $1/r$-Lipschitz on its domain. Define the function $f^*(x) \triangleq |f(x)|$. Note that for $\mu$ the uniform distribution on $\{z_1, \ldots, z_N\}$,

$$|\mathbb{E}[f(\mu)] - \mathbb{E}[f(\mathcal{D})]| = |\mathbb{E}[f(\mathcal{D})]| \geq 1 - N\alpha,$$

so we conclude that $W_1(\mu, \mathcal{D}) \geq r(1 - N\alpha)$. $\square$

**Lemma A.15** (Uniform distribution on box). *$I_d$ is $(N, 1/2)$-diverse for $N \leq \frac{1}{2}(d/18)^{d/2}$.*

*Proof.* By Fact A.3, $Q_{I_d}(r) \leq (18r^2/d)^{d/2}$. Taking $r = 1$ and applying Lemma A.14 allows us to conclude that $W_1(\mu, I_d) \geq 1 - N(18/d)^{d/2}$, from which the lemma follows. $\square$

As we stated in Lemma 2.3, a large family of pushforwards of the uniform distribution over $[0, 1]^d$ are similarly diverse. Below we restate this lemma and provide a complete proof.

**Lemma A.16** (Random expansive leaky ReLU networks). *Let $\gamma > 0$. For $k_0, \ldots, k_L \in \mathbb{N}$ satisfying $k_i \geq (1 + \gamma)k_{i-1}$ for all $i \in [L]$, let $\mathbf{W}_1 \in \mathbb{R}^{k_1 \times k_0}, \mathbf{W}_2 \in \mathbb{R}^{k_2 \times k_1}, \ldots, \mathbf{W}_L \in \mathbb{R}^{k_L \times k_{L-1}}$ be random weight matrices, where every entry of $\mathbf{W}_i$ is an independent draw from $\mathcal{N}(0, 1/k_i)$. For the function $F : \mathbb{R}^{k_0} \to \mathbb{R}^{k_L}$ given by*

$$F(x) \triangleq \mathbf{W}_L \psi_\lambda \left(\mathbf{W}_{L-1} \psi_\lambda \left(\cdots \psi_\lambda(\mathbf{W}_1 x) \cdots\right)\right),$$

*we have that $F(I_{k_0})$ is $(2^m, \beta)$-diverse for*

$$m = (k_0/2)\log(k_0/2) - k_L/\gamma - 1$$

$$\beta = \frac{1}{6}\left(\frac{\lambda}{2e^{1/\gamma}}\right)^L.$$

*So for example, if $\gamma, \lambda, L$ are constants, then $F(I_{k_0})$ is $(2^{\Omega(k_0 \log k_0)}, \Omega(1))$-diverse for $k_0$ sufficiently large.*

We will prove this inductively by first arguing that pushing anticoncentrated distributions through leaky ReLU (Lemma A.17) or through mildly "expansive" random linear functions (Lemma A.18) preserves anticoncentration to some extent:

**Lemma A.17.** *Let $0 < \lambda \le 1/2$. If a distribution $\mathcal{D}$ over $\mathbb{R}^d$ satisfies $Q_{\mathcal{D}}(r) \le \alpha$, then the pushforward $\mathcal{D}' \triangleq \psi_\lambda(\mathcal{D})$ also satisfies $Q_{\mathcal{D}'}(\lambda r) \le 2^d \alpha$, where here $\psi_\lambda(\cdots)$ denotes entrywise application of the leaky ReLU activation.*

*Proof.* Consider any ball $B(\nu, \lambda r)$ in $\mathbb{R}^d$. Take any orthant $K_S$ of $\mathbb{R}^d$, given by points whose $i$-th coordinates are nonnegative for $i \in S$ and negative for $i \notin S$. Let $B_S$ be the intersection of $B$ with this orthant. Then $\psi_\lambda^{-1}(B_S)$ consists of points $z \in K_S$ for which

$$\sum_{i \in S}(\lambda z_i - \nu_i)^2 + \sum_{i \notin S}((1-\lambda)z_i - \nu_i)^2 \le \lambda^2 r^2. \tag{3}$$

We can rewrite the left-hand side of (3) as

$$\lambda^2 \sum_{i \in S}(z_i - \nu_i \lambda)^2 + (1-\lambda)^2 \sum_{i \notin S}(z_i - \nu_i/(1-\lambda))^2 \ge \lambda^2 \|z - \nu(S)\|^2,$$

where in the last step we used $\lambda \le 1/2$ and define the vector $\nu^S \in \mathbb{R}^d$ by

$$\nu_i^S = \begin{cases} \nu_i^S/\lambda & i \in S \\ \nu_i^S/(1-\lambda) & i \notin S \end{cases}.$$

In other words, $\psi_\lambda^{-1}(B_S)$ is contained in $K_S \cap B(\nu(S), r)$. In particular,

$$\psi_\lambda^{-1}(B) \subset \bigcup_S K_S \cap B(\nu(S), r),$$

so $\mathbb{P}_{x \sim \mathcal{D}'}[x \in B] \le 2^d \cdot \alpha$ by a union bound. $\qquad\square$

**Lemma A.18.** *Suppose $n, d \in \mathbb{N}$ satisfy $n \ge (1+\gamma)d$ for some $\gamma > 0$. Let $\mathbf{W} \in \mathbb{R}^{n \times d}$ be a matrix whose entries are independent draws from $\mathcal{N}(0, 1/n)$. If a distribution $\mathcal{D}$ over $\mathbb{R}^d$ satisfies $Q_{\mathcal{D}}(r) \le \alpha$, then for the linear map $f : x \mapsto \mathbf{W}x$, the pushforward $\mathcal{D}' \triangleq f(\mathcal{D})$ satisfies $Q_{\mathcal{D}'}\left(\frac{\gamma r}{2(1+\gamma)}\right) \le \alpha$ with probability at least $1 - \exp(-\Omega(\gamma d))$.*

*Proof.* By Theorem A.6, for any $\epsilon > 0$ we have that $\sigma_{\min}(\mathbf{W}) \ge \epsilon \cdot \left(1 - \sqrt{\frac{d-1}{n}}\right)$ with probability at least $1 - (C\epsilon)^{n-d+1} - e^{-cn}$. Taking $\epsilon = 1/2C$ and noting that $1 - \sqrt{\frac{d-1}{n}} \ge \frac{\gamma}{1+\gamma}$, we conclude that

$$\mathbb{P}\left[\sigma_{\min}(\mathbf{W}) \ge \frac{\gamma}{2(1+\gamma)}\right] \ge 1 - \exp(-\Omega(\gamma d)).$$

Condition on this event. Now for any $\nu \in \mathbb{R}^n$, if we write $\nu$ as $\mathbf{W}\mu + \mu^\perp$ where $\mu^\perp$ is orthogonal to the column span of $\mathbf{W}$, then $\|\mathbf{W}x - \nu\|^2 = \|\mathbf{W}(x-\mu)\|^2 + \|\mu^\perp\|^2$. So $\|\mathbf{W}x - \nu\| \le \frac{\gamma r}{2(1+\gamma)}$ implies that $\|\mathbf{W}(x-\mu)\| \le \frac{\gamma r}{2(1+\gamma)}$. But because $\sigma_{\min}(\mathbf{W}) \ge \frac{\gamma}{2(1+\gamma)}$, we conclude that $\|x - \mu\| \le r$, from which the lemma follows. $\qquad\square$

We are now ready to prove Lemma 2.3:

*Proof of Lemma 2.3.* By Lemma A.14 it suffices to bound the Lévy concentration function. We will induct on the layers of $F$. For $i \in [L]$, let $F^{(i)}$ denote the sub-network

$$\mathbf{W}_i \psi_\lambda \left(\mathbf{W}_{L-1} \psi_\lambda \left(\cdots \psi_\lambda(\mathbf{W}_1 x) \cdots\right)\right),$$

and let $\mathcal{D}_i$ denote the pushforward $F^{(i)}(I_{k_0})$, which is a distribution over $\mathbb{R}^{k_i}$. We would like to apply Lemma A.18 to each of the weight matrices $\mathbf{W}_1, \ldots, \mathbf{W}_L$, so condition on the event that the lemma holds for these matrices, which happens with probability at least $1 - L\exp(-\Omega(\gamma d))$.

Recalling from Fact A.3 that $Q_{I_{k_0}}(r) \le (18r^2/k_0)^{k_0/2}$ for any $r > 0$, we get from Lemma A.18 applied to $\mathbf{W}_1$ that $Q_{\mathcal{D}_1}\left(\frac{\gamma r}{2(1+\gamma)}\right) \le (18r^2/k_0)^{k_0/2}$.

Suppose inductively that we have shown that $Q_{\mathcal{D}_i}(r_i) \leq \alpha_i$ for some $r_i, \alpha > 0$. Then by Lemma A.17 and Lemma A.18 applied to weight matrix $\mathbf{W}_{i+1}$, we conclude that

$$Q_{\mathcal{D}_{i+1}}(r_{i+1}) \leq \alpha_{i+1} \quad \text{for} \quad r_{i+1} = \frac{\lambda \gamma r_i}{2(1+\gamma)}, \alpha_{i+1} = 2^{k_i} \alpha_i. \tag{4}$$

Unrolling the recursion (4), we conclude that $Q_{\mathcal{D}_L}(r_L) \leq \alpha_L$ for

$$r_L = r\lambda^{L-1} \left( \frac{\gamma}{2(1+\gamma)} \right)^L \geq r \cdot \left( \frac{\lambda \gamma}{2(1+\gamma)} \right)^L \geq r \cdot \left( \frac{\lambda}{2 \cdot e^{1/\gamma}} \right)^L$$

$$\alpha_L = 2^{k_1 + \cdots + k_{L-1}} (18r^2/k_0)^{k_0/2} \leq 2^{k_L/\gamma} (18r^2/k_0)^{k_0/2}, \tag{5}$$

where the inequality in (5) follows from the fact that $k_1 + \cdots + k_{L-1} \leq k_{L-1}(1 + 1/\gamma) \leq k_L/\gamma$. By Lemma A.14, $F(I_{k_0}) = \mathcal{D}_L$ is $(N, r_L(1 - N\alpha_L))$-diverse. The lemma follows by taking $r = 1/3$ and $2^m = N = 1/2\alpha_L$. □

# B    DEFERRED PROOFS FROM SECTION 3

## B.1    PROOF OF COROLLARY 3.3

Corollary 3.3 is an immediate consequence of the following which appeared in (Chen et al., 2020b):

**Lemma B.1.** *For any function $P : \{\pm 1\}^k \to \{\pm 1\}$, there is a collection of $k$ weight matrices $\mathbf{W}_1, \ldots, \mathbf{W}_k$ with entries in $\mathbb{R}_{O(k)}$ for which*

$$P(x) = \mathbf{W}_k \phi(\cdots \phi(\mathbf{W}_1 x) \cdots) \tag{6}$$

*for all $x \in \{\pm 1\}^k$, and for which $\|\mathbf{W}_i\| \leq O(1)$. Furthermore, the size of the network on the right-hand side of (6) is at most $O(2^k \cdot k)$.*

We give a proof for completeness to make explicit the dependence of the parameters on $k$.

*Proof.* Consider the Fourier expansion $F(x) = \sum_{S \subseteq [k]} \widehat{F}[S] \prod_{i \in S} x_i$. We show how to represent each Fourier basis function $\prod_{i \in S} x_i$ as a ReLU network with at most $k$ layers. Observe that for any $x_1, x_2 \in \{\pm 1\}$,

$$x_1 \cdot x_2 = \phi(x_1 + x_2) + \phi(-x_1 - x_2) - \phi(x_2) - \phi(-x_2), \tag{7}$$

which is a two-layer neural network of size 4 whose two weight matrices have operator norm at most 3. Suppose inductively that for some $1 \leq m < n$, there exist weight matrices $\mathbf{W}'_1, \ldots, \mathbf{W}'_m$ for which $\prod_{i=1}^m x_i = \mathbf{W}'_m \phi(\cdots \phi(\mathbf{W}'_1 x) \cdots)$ for all $x \in \{\pm 1\}^k$, that this network has size $4m$, and that $\prod_{i=1}^m \|\mathbf{W}'_i\| \leq 6^m$.

We now show how to compute $\prod_{i=1}^{m+1} x_i$. Define $\mathbf{W}''_1$ by adding the $m$-th standard basis vector as a new row at the bottom of $\mathbf{W}'_1$. For every $1 < i \leq m$, define $\mathbf{W}''_i$ to be the matrix given by appending a column of zeros to the right of $\mathbf{W}''_i$ and then a new row at the bottom consisting of zeros except in the rightmost entry. Note that $\|\mathbf{W}''\|_1 = \max(1, \|\mathbf{W}'_i\|)$. Define the network $F_m : \mathbb{R}^k \to \mathbb{R}^2$ by $F_m(x) = \mathbf{W}''_m \phi(\cdots \phi(\mathbf{W}''_1 x) \cdots)$.

Letting $v, e \in \mathbb{R}^2$ be the vectors $(1, 1)$ and $(0, 1)$, we can use (7) to conclude that

$$\prod_{i=1}^{m+1} x_i = \phi \left( \prod_{i=1}^m x_i + x_{m+1} \right) + \phi \left( -\prod_{i=1}^m x_i - x_{m+1} \right) - \phi(x_{m+1}) - \phi(-x_{m+1})$$

$$= \phi(v^\top F_m(x)) + \phi(-v^\top F_m(x)) - \phi(e^\top F_m(x)) - \phi(-e^\top F_m(x)).$$

We can thus write $\prod_{i=1}^{m+1} x_i$ as the ReLU network

$$\prod_{i=1}^{m+1} x_i = \mathbf{W}'''_{m+1} \phi(\cdots \phi(\mathbf{W}'''_1 x) \cdots) \tag{8}$$

where

$$\mathbf{W}_{m+1}''' = (1, 1, -1, -1), \mathbf{W}_m''' = \begin{pmatrix} v^\top \mathbf{W}_m'' \\ -v^\top \mathbf{W}_m'' \\ e^\top \mathbf{W}_m'' \\ -e^\top \mathbf{W}_m'' \end{pmatrix}, \mathbf{W}_i''' = \mathbf{W}_i'' \quad \forall 1 \leq i < m.$$

Note that the entries of any $\mathbf{W}_i'''$ are in $\{0, \pm1\}$ and thus have bit complexity at most 2. Additionally, $\|\mathbf{W}_{m+1}'''\| \leq 2$, $\mathbf{W}_m''' \leq 3\|\mathbf{W}_m''\| = 3 \cdot \max(1, \|\mathbf{W}_m'\|)$, and $\|\mathbf{W}_i''\| = \max(1, \|\mathbf{W}_i'\|)$ for all $1 \leq i < m$, so $\prod_{i=1}^{m+1} \|\mathbf{W}_i'''\| \leq 6^{m+1}$. Furthermore, the size of the network in (8) is $4m + 4$. This completes the inductive step and we conclude that any Fourier basis function $\prod_{i \in S} x_i$ can be implemented by an $|S|$-layer ReLU network with size $4|S|$ and the product of whose weight matrices' operator norms is at most $6^{|S|}$.

In particular, as the biases in the network are zero, we can rescale the weight matrices so they have equal operator norm, in which case they each have operator norm at most $O(1)$ and entries in $\mathbb{R}_{O(k)}$

Finally note that because the Fourier coefficients are given by $\mathbb{E}[F(x) \prod_{i \in S} x_i]$, they are all multiples of $1/2^k$ and thus have bit complexity $O(k)$. The proof follows from applying Lemma B.2 to these Fourier basis functions and $\lambda$ given by the Fourier coefficients of $P$, as $\|\lambda\| = \|P\| = 1$. $\qquad\square$

The above proof required the following basic fact:

**Lemma B.2.** *Let $\tau, \tau' \in \mathbb{N}$, and let $\lambda \in \mathbb{R}_\tau^r$. Given neural networks $F_1, \ldots, F_r : \mathbb{R}^d \to \mathbb{R}$ each with $L$ layers and whose weight matrices $\{\mathbf{W}_i^{(1)}\}, \ldots, \{\mathbf{W}_i^{(r)}\}$ have operator norm bounded by some $R > 0$ and entries in $\mathbb{R}_{\tau'}$, their linear combination $\sum_i \lambda_i F_i$ is a neural network with $L$ layers, size given by the sum of the sizes of $F_1, \ldots, F_r$, and weight matrices $\mathbf{W}_1, \ldots, \mathbf{W}_L$ with entries in $\mathbb{R}_{O(\tau+\tau')}$ and satisfying $\|\mathbf{W}_1\| \leq R\sqrt{r}$, $\|\mathbf{W}_L\| \leq R\|\lambda\|$, and $\|\mathbf{W}_i\| \leq R$ for all $1 < i < L$. Here $\lambda \in \mathbb{R}^r$ is the vector with entries $\lambda_i$.*

*Proof.* Denote the $i$-th weight matrix of $F_j$ by $\mathbf{W}_i^{(j)}$. Define $\mathbf{W}_1$ to be the vertical concatenation of $\mathbf{W}_1^{(1)}, \ldots, \mathbf{W}_1^{(r)}$, and for every $1 < i < L$, define $\mathbf{W}_i$ to be the block diagonal concatenation of $\mathbf{W}_i^{(1)}, \ldots, \mathbf{W}_i^{(r)}$. Finally, define $\mathbf{W}_L$ to be the row vector given by the product

$$\lambda^\top \begin{pmatrix} \mathbf{W}_L^{(1)} & 0 & \cdots & 0 \\ 0 & \mathbf{W}_L^{(2)} & \cdots & 0 \\ \vdots & 0 & \ddots & 0 \\ 0 & 0 & \cdots & \mathbf{W}_L^{(r)} \end{pmatrix}$$

For all $1 < i < L$, $\|\mathbf{W}_i\| \leq \max_{j \in [r]} \|\mathbf{W}^{(i)}\|$, and additionally $\|\mathbf{W}_1\|^2 \leq \sum_{j=1}^r \|\mathbf{W}_1^{(j)}\|^2$ and $\|\mathbf{W}_L\| \leq \|\lambda\| \max_{j \in [r]} \|\mathbf{W}_L^{(j)}\|$. $\qquad\square$

### B.2 PROOF OF LEMMA 3.4

We will need the following helper lemma about means of truncations of sub-Gaussian random variables:

**Lemma B.3.** *If $Z$ is $\sigma^2$-sub-Gaussian and mean zero, then for any interval $I = [a, b]$ with $a \leq 0 \leq b$, we have $|\mathbb{E}[Z \cdot \mathbb{1}[Z \notin I]]| \leq O(b - a + \sigma) \cdot \exp(-\min(-a, b)^2 / 2\sigma^2)$.*

*Proof.* Define the random variable $Z' = Z \cdot \mathbb{1}[Z \notin I]$. Then by integration by parts,

$$\mathbb{E}[Z'] \leq \mathbb{E}[Z \cdot \mathbb{1}[Z > b]]$$
$$= \int_0^\infty \mathbb{P}[Z' > t]dt$$
$$= b\,\mathbb{P}[Z > b] + \int_b^\infty \mathbb{P}[Z > t]dt$$
$$\leq b\exp(-b^2/2\sigma^2) + O(\sigma \cdot \exp(-b^2/2\sigma^2))$$
$$\leq O(b+\sigma) \cdot \exp(-b^2/2\sigma^2).$$

and similarly, $\mathbb{E}[Z'] \geq \mathbb{E}[Z \cdot \mathbb{1}[Z < -a]] \geq O(a-\sigma) \cdot \exp(-b^2/2\sigma^2)$, completing the proof. □

We now complete the proof of Lemma 3.4.

*Proof of Lemma 3.4.* Without loss of generality we can assume that $\mathbb{E}[X] = 0$ and $\mathbb{E}[Y] = \alpha$. If $\alpha \geq c\sigma$ for some sufficiently large absolute constant, then we can simply take $t = \alpha/2$ and get that $|\mathbb{P}[X > t] - \mathbb{P}[Y > t]| \geq 1/2$. Now suppose $\alpha < c\sigma$, and let $I = [-r, r+\alpha]$ for $r = \sigma\sqrt{\log(C\sigma/\alpha)}$ for some large constant $C > 0$. Note that by this choice of $r$,

$$r\exp(-r^2/2\sigma^2) \leq O(\alpha),$$

where the constant factor can be made arbitrarily small by picking $C$ sufficiently lage. Define the random variables $X' \triangleq X \cdot \mathbb{1}[X \in I]$ and $Y' \triangleq Y \cdot \mathbb{1}[Y \in I]$. Then

$$\alpha = \mathbb{E}[Y] - \mathbb{E}[X] = \mathbb{E}[Y'] - \mathbb{E}[X'] + \mathbb{E}[Y \cdot \mathbb{1}[Y \notin I]] - \mathbb{E}[X \cdot \mathbb{1}[X \notin I]]. \tag{9}$$

By Lemma B.3,

$$\mathbb{E}[X \cdot \mathbb{1}[X \notin I]] \leq O(2r + \alpha + \sigma) \cdot \exp(-(r+\alpha)^2/2\sigma^2) \leq O(r) \cdot \exp(-r^2/2\sigma^2) \leq O(\alpha) \tag{10}$$

and similarly

$$\mathbb{E}[Y \cdot \mathbb{1}[Y \notin I]] \leq \mathbb{E}[Y] \cdot \mathbb{P}[Y \notin I] + O(2r + \alpha + \sigma) \cdot \exp(-(r+\alpha)^2/2\sigma^2).$$
$$\leq 2\alpha\exp(-r^2/2\sigma^2) + O(r) \cdot \exp(-(r+\alpha)^2/2\sigma^2) \leq O(\alpha). \tag{11}$$

Additionally, we have

$$\mathbb{E}[X'] - \mathbb{E}[Y'] = \int_0^{\alpha+r}(\Phi_{Y'}(z) - \Phi_{X'}(z))dz - \int_{-\alpha}^0 (\Phi_{X'}(z) - \Phi_{Y'}(z))dz \tag{12}$$

where $\Phi_Z(z)$ denotes the cdf at $z$ of random variable $Z$. Putting (9), (10), (11), (12) together, we conclude that

$$\min\left(\int_0^{\alpha+r}(\Phi_{Y'}(z) - \Phi_{X'}(z))dz, \int_{-\alpha}^0 (\Phi_{X'}(z) - \Phi_{Y'}(z))dz\right) \geq \Omega(\alpha),$$

where the constant factor can be made arbitrarily close to 1/2 by making $C$ sufficiently small. By averaging, we conclude that there exists $t \in [-\alpha, \alpha + r]$ for which

$$|\mathbb{P}[X' > t] - \mathbb{P}[Y' > t]| \geq \Omega(\alpha/r).$$

But $\mathbb{P}[X \notin I], \mathbb{P}[Y \notin I] \leq O(\exp(-r^2/2\sigma^2)) \leq O(\alpha/r)$, where the absolute constant can be made arbitrarily small by making $C$ sufficiently small. The claim follows by a union bound, recalling the definition of $X', Y'$. □

### B.3 Thresholds of Networks as Circuits

In the proof of Theorem 3.2, we also need the following basic fact that signs of ReLU networks can be computed in P/poly.

**Lemma B.4.** *For any $f \in \mathcal{F}^*$, there is a Turing machine that, given any input $y$, outputs $\mathrm{sgn}(f(y))$ after $\mathrm{poly}(d)$ steps.*

*Proof.* Recall that the weight matrices $\mathbf{W}_1, \ldots, \mathbf{W}_L$ of $f$ have entries in $\mathbb{R}_\tau$ for $\tau = \mathrm{poly}(d)$. So for any $1 \le \ell \le L$, diagonal matrices $\mathbf{D}_1 \in \{0,1\}^{k_1 \times k_1}, \ldots, \mathbf{D}_{\ell-1} \in \{0,1\}^{k_{\ell-1} \times k_{\ell-1}}$, and vector $y \in \{\pm 1\}^d$, every entry of the vector

$$\mathbf{W}_\ell \mathbf{D}_{\ell-1}(\mathbf{W}_{\ell-1}\mathbf{D}_{\ell-2}(\cdots(\mathbf{W}_1 y + b_1)\cdots) + b_{\ell-1}) + b_\ell$$

has bit complexity bounded by

$$\log_2\left(\ell \cdot 2^{O(\ell\tau)} \prod_{i=1}^{\ell-1} k_i\right) = O(\ell\tau + S) = \mathrm{poly}(d),$$

where in the second step we used that $\log(k_i) \le k_i$ for all $i \in [\ell-1]$. So for any input to $f$, every intermediate activation has $\mathrm{poly}(d)$ bit complexity.

The Turing machine we exhibit for computing $\mathrm{sgn}(f(y))$ will compute the activations in the network layer by layer. The entries of $\mathbf{W}_1 y + b_1$ can readily be computed in $\mathrm{poly}(d)$ time. Now given the vector of activations

$$v = \mathbf{W}_\ell \phi(\cdots \phi(\mathbf{W}_1 y + b_1)\cdots) + b_\ell$$

for some $\ell \ge 1$ (where $v$ is represented on a tape of the Turing machine as a bitstring of length $\mathrm{poly}(d)$), we need to compute $\mathbf{W}_{\ell+1}\phi(v) + b_{\ell+1}$. The ReLU activation can be readily computed in $\mathrm{poly}(d)$ time, so in $\mathrm{poly}(d)$ additional steps we can form this new vector of activations at the $(\ell+1)$-layer. So within $S \cdot \mathrm{poly}(d) = \mathrm{poly}(d)$ steps the Turing machine will have written down $f(y)$ (represented as a bitstring of length $\mathrm{poly}(d)$) on one of its tapes, after which it will return the sign of this quantity. $\square$

### B.4 PROOF OF THEOREM 3.2

We now give a complete proof of Theorem 3.2:

*Proof.* The parameter $m$ will be clear from context in the following discussion, so for convenience we will refer to $d(m)$ and $G_m$ as $d$ and $G$. Let $k, P, G$ be such that the outcome of Assumption 1 holds, and $\mathrm{negl}(\cdot)$ denote the function indicating the extent to which $G$ fools poly-sized circuits. By Corollary 3.3, every output coordinate of $G$ is computable by a network in $\mathcal{C}_{L,S,m}^{\tau,\Lambda}$ for $\tau = O(k), \Lambda = \exp(O(k)), L = k, S = O(2^k k)$.

We first check that $W_1(G(U_m), U_d) > 1/3$. Note that $G(U_m)$ has support of size $2^m$. In Lemma A.13 we can take $\mu = U_d$ and conclude that $\mu$ is $(2^m, 2(1 - 2^{m-d}))$-diverse, so $W_1(G(U_m), U_d) \ge 2(1 - 2^{m-d}) = 2(1 - 2^{m-m^c}) \ge 1$.

It remains to check that $G$ fools $\mathcal{F}^*$ relative to $U_d$. Suppose to the contrary that there exists some $f \in \mathcal{F}^*$ and absolute constant $a > 0$ for which $|\mathbb{E}[f(G(U_m))] - \mathbb{E}[f(U_d)]| > 1/d^a$. We will argue that this implies there is a poly-sized circuit $C : \{\pm 1\}^d \to \{\pm 1\}$ distinguishing $G(U_m)$ from $U_d$.

First note that for any threshold $t \in \mathbb{R}_\tau$, by Lemma B.4 there is a Turing machine $\mathcal{M}_\tau : \{\pm 1\}^d \to \{\pm 1\}$ that computes $y \mapsto \mathrm{sgn}(f(y) - t)$ with $\tau$ bits of advice. So if there existed a threshold $t \in \mathbb{R}_\tau$ for which

$$|\mathbb{E}[\mathcal{M}_\tau(G(U_m))] - \mathbb{E}[\mathcal{M}_\tau(U_d)]| > 1/d^{a'}, \tag{13}$$

for some constant $a' > 0$, then by Fact A.8, there would exist a Boolean circuit $C$ distinguishing $G(U_m)$ from $U_d$ with non-negligible advantage, contradicting Assumption 1 and concluding the proof.

We will apply Lemma 3.4 to show the existence of such a threshold $t$. Specifically, define random variables $X = f(G(U_m))$ and $Y = f(U_d)$. By Corollary A.5 applied to the $\mathrm{poly}(d)$-Lipschitz function $f : \{\pm 1\}^d \to \{\pm 1\}$, $Y - \mathbb{E}[Y]$ is $\mathrm{poly}(d)$-sub-Gaussian. And recalling that $G \in \mathcal{C}_{L,S,m}^{\tau,\Lambda}$ for $\Lambda = \exp(O(k))$, we can apply Corollary A.5 to the $\mathrm{poly}(d) \cdot O_k(1)$-Lipschitz function $f \circ G : \{\pm 1\}^m \to \{\pm 1\}$ to conclude that $X - \mathbb{E}[X]$ is $\sigma^2$-sub-Gaussian for $\sigma \triangleq \mathrm{poly}(m) \cdot \exp(O(k)) = \mathrm{poly}(d)$. By Lemma 3.4, there exists a threshold $t$ for which the left-hand side of (13) exceeds $\min(1/2, \widetilde{\Omega}(n^{-a}/\sigma))$, which is not negligible.

It remains to verify that $t$ has bit complexity at most $\mathrm{poly}(d)$. As the entries in the weight matrices and biases in $f$ all have bit complexity $\mathrm{poly}(d)$ and $f$ has size and depth $\mathrm{poly}(d)$, $f(y)$ has bit

complexity $\text{poly}(d)$ for any $y \in \{\pm 1\}^d$. Similarly, the entries in the weight matrices and biases in $G$ all have bit complexity $O(k) = O(1)$, so $f(G(x))$ has bit complexity $\text{poly}(d)$ for any $x \in \{\pm 1\}^m$. By the bound on $t$ in Lemma 3.4 and our bound on $\sigma$ above, $t$ therefore also has $\text{poly}(d)$ bit complexity. $\qquad \square$

## B.5 Proof of Lemma 3.6

*Proof.* Suppose to the contrary that there existed some function $f \in \mathcal{F}'$ for which

$$|\mathbb{E}[f(J(p))] - \mathbb{E}[f(J(q))]| > 2\epsilon \cdot \Lambda'.$$

By Lemma 2.1 and our choice of $S''$, the composition $f \circ J : \mathbb{R}^s \to \mathbb{R}$ can be computed by a network in $\mathcal{C}_{L,S,s}^{\tau, \Lambda\Lambda'\sqrt{r}}$ whose bias and weight vector entries in the output layer lie in $\mathbb{R}_{\tau'}$.

We first show why this would lead to a contradiction. Consider the function $h \triangleq \frac{1}{C} \cdot f \circ J$ for

$$C = 2^{\lceil \log_2 \Lambda' \sqrt{r} \rceil} \in [\Lambda', 2\Lambda'),$$

which can be computed by taking the network computing $f \circ J$ and scaling the bias and weight vector in the output layer by $C$. Note that this scaling results in bias and weight vector entries in the output layer for $h$ with bit complexity $\tau' + \lceil \log_2 \Lambda' \sqrt{r} \rceil = \tau$. Furthermore, $h$ is $\Lambda\Lambda'\sqrt{r}/C \leq \Lambda$-Lipschitz, so $h \in \mathcal{C}_{L,S,s}^{\tau, \Lambda}$. On the other hand, we would have

$$|\mathbb{E}[h(p)] - \mathbb{E}[h(q)]| > 2\epsilon \Lambda' \sqrt{r}/C \geq \epsilon,$$

yielding the desired contradiction of the assumption that $W_{\mathcal{F}}(p,q) \leq \epsilon$. $\qquad \square$

## B.6 Proof of Fact 3.7

*Proof.* Note that $h(U_n)$ is the uniform distribution over multiples of $1/2^n$ in the interval $[0,1)$. Given any such multiple $z$, let $p_z$ denote the uniform distribution over $[z, z + 1/2^n)$. One way of sampling from $I_1$ is thus to sample $z$ from $h(U_n)$ and then sample from $p_z$.

Now consider any 1-Lipschitz function $f : \mathbb{R} \to \mathbb{R}$. Note that for any $z'$ in the support of $p_z$, $|f(z) - f(z')| \leq 1/2^n \leq \epsilon$. We have

$$|\mathbb{E}[f(h(U_n))] - \mathbb{E}[f(I_1)]| = \left| \mathbb{E}_{z \sim h(U_n)} \left[ \mathbb{E}_{z' \sim p_z} [f(z) - f(z')] \right] \right| \leq \epsilon$$

as desired. $\qquad \square$

## B.7 Proof of Lemma 3.8

*Proof.* Let $n \triangleq \lceil \log(1/\epsilon) \rceil$. For every $i \in [r]$, define $S_i \triangleq \{(i-1) \cdot n + 1, \dots, i \cdot n\}$. Take $J$ to be the linear function where for every $i \in [r]$, the $i$-th output coordinate of $J$ is the linear function which maps $y \in \mathbb{R}^s$ to $\langle w_i, y + \mathbf{1} \rangle$ where $w_i$ is zero outside of $S_i$ and, over coordinates indexed by $S_i$, equal to the vector $(1/4, \dots, 1/2^{n+1})$. Note that each output coordinate of $J$ is computed by a function in $\mathcal{C}_{1,0,s}^{n+1,O(1)}$.

By Fact 3.7 and Fact A.11,

$$W_{\mathcal{F}^*}(J(U_s), I_r) \leq \text{poly}(r) \cdot W_1(J(U_s), I_r) \leq \text{poly}(r) \cdot \epsilon.$$

On the other hand, by Lemma 3.6 and the fact that the union of $\mathcal{C}_{L-1,S-r,r}^{\tau - \lceil \log_2 \Lambda \sqrt{r} \rceil, \Lambda}$ over $\tau, \Lambda, L, S = \text{poly}(s)$ is still $\mathcal{F}^*$, we conclude that

$$W_{\mathcal{F}^*}(J(\widetilde{\mathcal{D}}), J(U_s)) \leq O(\epsilon\sqrt{r}),$$

from which the lemma follows by triangle inequality. $\qquad \square$

## B.8 PROOF OF LEMMA 3.9

*Proof.* Let $J : \mathbb{R}^s \to \mathbb{R}^r$ be given by Lemma 3.8. We know that $W_{\mathcal{F}^*}(J(\widetilde{\mathcal{D}}), I_r) \le \epsilon \cdot \mathrm{poly}(r)$. By Lemma 3.6 applied to these two distributions and the generator function $H$, together with the fact that the union of $\mathcal{C}_{L-L',S-d(S'+1),d}^{\tau'-\lceil \log_2 \Lambda \sqrt{d} \rceil, \Lambda}$ over $\Lambda, L, S = \mathrm{poly}(s)$ is still $\mathcal{F}^*$, we thus have that $W_{\mathcal{F}^*}(H(J(\widetilde{\mathcal{D}})), H(I_r)) \le O(\epsilon \Lambda' \sqrt{d}) \cdot \mathrm{poly}(r) = \epsilon \Lambda' \cdot \mathrm{poly}(s)$.

We will thus take $J'$ in the lemma to be $H \circ J$. By Lemma 2.1, each output coordinate of $J'$ is computed by a function in $\mathcal{C}_{L'+1,r+S',s}^{\max(O(\log(1/\epsilon)),\tau'),O(\Lambda'\sqrt{d})}$ as claimed. $\qquad\square$

## B.9 PROOF OF THEOREM 3.5

*Proof of Theorem 3.5.* The parameter $m$ will be clear from context in the following discussion, so for convenience we will refer to $r(m), d(m), \epsilon(m), H_m, G_m$ as $r, d, \epsilon, H, G$, and similarly for the network parameters $\tau', \Lambda', L', S'$.

It is easy to verify condition 3 before we even define $G$: because $G(U_m)$ is a uniform distribution on $2^m$ points (with multiplicity) and $H(I_r)$ is $(2^m, \Omega(1))$-diverse, $W_1(G(U_m), I_d) \ge \Omega(1)$ as claimed.

Let $s = r \cdot \lceil \log(1/\epsilon) \rceil$. As we are assuming $\epsilon \ge \exp(-O(m))$, $s \le r \cdot m = m^c$ for some constant $c > 1$. If $s \le m$, then define $G' : \mathbb{R}^m \to \mathbb{R}^s$ to be the map given by projecting to the first $s$ coordinates so that $G'(U_m)$ and $U_s$ are identical as distributions. Otherwise, take $G'$ to be the generator $G : \mathbb{R}^m \to \mathbb{R}^s$ constructed in Theorem 3.2, recalling that $W_{\mathcal{F}^*}(G'(U_m), U_s) \le \mathrm{negl}(m) \le \epsilon$.

Next, by applying Lemma 3.9 to $\widetilde{\mathcal{D}} = G'(U_m)$, we get a function $J' : \mathbb{R}^s \to \mathbb{R}^d$ each of whose output coordinates is computed by a function in $\mathcal{C}_{L'+1,r+S',s}^{\max(O(\log(1/\epsilon)),\tau'),O(\Lambda'\sqrt{d})}$ such that

$$W_{F_d}(J'(G'(U_m)), H(I_r)) \le \epsilon \Lambda' \cdot \mathrm{poly}(m) \le \epsilon \cdot \mathrm{poly}(m), \tag{14}$$

where the second step follows by our assumption on $\Lambda'$.

We will take $G \triangleq J' \circ G'$. (14) establishes condition 2 of the theorem. Finally, by Lemma 2.1, every output coordinate of $G$ can be realized by a network in $\mathcal{C}_{L,S,m}^{\tau,\Lambda}$ for $\tau = \max(O(\log(1/\epsilon)), \tau', O(1))$, $\Lambda = O(\Lambda'\sqrt{ds}) = O(\Lambda') \cdot \mathrm{poly}(m)$, $L = L' + O(1)$, and $S = O(s) + r + S' = O(s) + S'$ (where we used the fact that $r = s/\lceil \log(1/\epsilon) \rceil < s$). This establishes condition 1 of the theorem. $\qquad\square$

## B.10 PROOF OF LEMMA 3.10

*Proof.* Note that

$$h_\xi(x) = \phi(x/\xi + 1) - \phi(x/\xi - 1) - 1, \tag{15}$$

so we can take weight matrices

$$\mathbf{W}_1 = \begin{pmatrix} 1/\xi \\ 1/\xi \end{pmatrix} \qquad \mathbf{W}_2 = (1 \quad -1)$$

and biases $b_1 = (1, -1)$ and $b_2 = -1$. Note that $h_\xi$ is $1/\xi$-Lipschitz. We conclude that $h_\xi \in \mathcal{C}_{2,2,1}^{\tau,1/\xi}$. $\qquad\square$

## B.11 PROOF OF LEMMA 3.11

*Proof.* We first verify that $W_1(G_0(U_m), G(\gamma_m)) \le \epsilon$. Take any 1-Lipschitz function $f$. Note that we can sample from $U_m$ by sampling a vector $g$ from $\gamma_m$, applying $h_\xi$ entrywise to $g$, and replacing each resulting entry of $h_\xi(g)$ by its sign; importantly, the last step only affects entries $i \in [m]$ for which $|g_i| < \xi$.

We will define $\mathcal{E}$ to be the event that $|g_i| \ge \xi$ for all $i \in [m]$, noting that

$$\mathbb{P}[\mathcal{E}] \ge 1 - m \cdot \mathop{\mathbb{P}}_{g \sim \mathcal{N}(0,1)}[|g| < \xi] \ge 1 - m\xi\sqrt{2/\pi}.$$

We can thus write

$$|\mathbb{E}[f(G_0(U_m))] - \mathbb{E}[f(G(\gamma_m))]|$$

$$= \left| \mathop{\mathbb{E}}_{g \sim \gamma_m} [(f(G_0(h_\xi(g))) - f(G(g))) \cdot \mathbb{1}[\mathcal{E}] + (f(G_0(\mathrm{sgn}(h_\xi(g)))) - f(G(g))) \cdot \mathbb{1}[\mathcal{E}^c]] \right|$$

$$= \left| \mathop{\mathbb{E}}_{g \sim \gamma_m} [(f(G_0(\mathrm{sgn}(h_\xi(g)))) - f(G_0(h_\xi(g)))) \cdot \mathbb{1}[\mathcal{E}^c]] \right|. \tag{16}$$

By Fact A.2, $f \circ G_0$ is $\Lambda'' \sqrt{d}$-Lipschitz. Furthermore, because $h_\xi(g) \in [-1, 1]^m$, $\|\mathrm{sgn}(h_\xi(g)) - h_\xi(g)\| \le \sqrt{m}$. We can thus upper bound (16) by

$$\le \Lambda'' \sqrt{md} \cdot \mathbb{P}[\mathcal{E}^c] \le \Lambda'' \xi \sqrt{(2/\pi) m^3 d} \le \epsilon$$

so $W_1(G_0(U_m), G(\gamma_m)) \le \epsilon$ as desired.

It remains to bound the complexity of $G$. For any $i \in [d]$, we can apply Lemma 2.1 with $f$ given by the $i$-th output coordinate of $G_0$ and $J$ given by the map which applies $h_\xi$ to every entry of the input. We thus conclude that $G \in \mathcal{C}_{L,S,m}^{\tau,\Lambda}$ for $\tau = \max(\tau'', \log_2(1/\xi)) = \max(\tau'', O(\log(\Lambda'' md/\epsilon)))$, $\Lambda = \Lambda'' \sqrt{m}/\xi = O(\Lambda''^2 m^2 \sqrt{d}/\epsilon)$, $L = L'' + 2$, $S = 3m + S''$ as claimed. $\qquad \square$

## C  FOOLING ReLU NETWORKS WOULD IMPLY NEW CIRCUIT LOWER BOUNDS

In this section we show that even exhibiting generators with logarithmic stretch that can fool all ReLU network discriminators of constant depth and slightly superlinear size would yield breakthrough circuit lower bounds.

First, in Section C.1 we review basics about average-case hardness and recall the state-of-the-art for lower bounds against $\mathsf{TC}^0$. Then in Section C.2 we present and prove the main result of this section, Theorem C.2.

### C.1  AVERAGE-CASE HARDNESS AND $\mathsf{TC}^0$

One of the most common notions of hardness for a class of functions $\mathcal{F}$ is *worst-case hardness*, that is, the existence of functions which cannot be computed by functions in $\mathcal{F}$.

**Definition 10** (Worst-case hardness). *Given a class of Boolean functions $\mathcal{F}$, a sequence of functions $f_n : \{\pm 1\}^n \to \{\pm 1\}$ is* worst-case-hard *for $\mathcal{F}$ if for every $f : \{\pm 1\}^n \to \{\pm 1\}$ in $\mathcal{F}$, there is some input $x \in \{\pm 1\}^n$ for which $f(x) \ne f_n(x)$.*

A more robust notion of hardness is that of *average-case hardness*, which implies worst-case hardness. For any $f_n \in \mathcal{F}$, rather than simply require that there is *some* input on which $f$ and $f_n$ disagree, we would like that over some fixed distribution over possible inputs, the probability that $f$ and $f_n$ output the same value is small. Typically, this fixed distribution is the uniform distribution over $\{\pm 1\}^n$, but in many situations even showing average-case hardness with respect to less natural distributions is open.

**Definition 11** (Average-case hardness). *Given a class of Boolean functions $\mathcal{F}$, a function $\epsilon : \mathbb{N} \to [0, 1/2)$, and a sequence of distributions $\{\mathcal{D}_n\}_n$ over $\{\pm 1\}^n$, a sequence of functions $f_n : \{\pm 1\}^n \to \{\pm 1\}$ is $(1/2 + \epsilon(n))$-average-case-hard for $\mathcal{F}$ with respect to $\{\mathcal{D}_n\}$ if for every $f : \{\pm 1\}^n \to \{\pm 1\}$ in $\mathcal{F}$,*

$$\mathop{\mathbb{P}}_{x \sim \mathcal{D}_n} [f(x) = f_n(x)] \le \frac{1}{2} + \epsilon(n).$$

By a counting argument, for any reasonably constrained class $\mathcal{F}$ there must *exist* functions which are worst/average-case hard for $\mathcal{F}$. A central challenge in complexity theory has been to exhibit *explicit* hard functions for natural complexity classes. In the context of this work, by explicit we simply mean that there is a polynomial-time algorithm for evaluating the function.

The complexity class we will focus on in this section is $\mathsf{TC}^0$, the class of constant-depth linear threshold circuits of polynomial size:

**Definition 12** (Linear threshold circuits). *A linear threshold circuit of size $S$ and depth $D$ is any Boolean circuit of size $S$ and depth $D$ whose gates come from the set $G$ of all linear threshold functions mapping $x \in \{\pm 1\}^n$ to $\mathrm{sgn}(\langle w, x\rangle - b)$ for some arity $n \in \mathbb{N}$, vector $w \in \mathbb{R}^n$, and bias $b \in \mathbb{R}$. $\mathsf{TC}^0$ is the set of all linear threshold circuits of size $\mathrm{poly}(n)$ and depth $O(1)$.*[6]

The best-known worst-case hardness result for $\mathsf{TC}^0$ is that of (Impagliazzo et al., 1997) who showed:

**Theorem C.1** ((Impagliazzo et al., 1997)). *Let $f_n : \{\pm 1\}^n \to \{\pm 1\}$ be the parity function on $n$ bits. For any depth $D \geq 1$, any linear threshold circuit of depth $D$ must have at least $n^{1+c\theta^{-D}}$ wires, where $c > 0$ and $\theta > 1$ are absolute constants.*

In (Chen et al., 2016) this worst-case hardness result was upgraded to an average-case hardness result with respect to the uniform distribution over the hypercube. Remarkably, this slightly superlinear lower bound from (Impagliazzo et al., 1997) has not been improved upon in over two decades!

We remark that the discussion about lower bounds for threshold circuits is a very limited snapshot of a rich line of work over many decades. We refer to the introduction in (Chen & Tell, 2019) for a more detailed overview of this literature.

## C.2 Hardness Versus Randomness for GANs

For convenience, given sequences of parameters $D(m), S(m) \in \mathbb{N}$ (these will eventually correspond to the depth and size of the linear threshold circuits against which we wish to show lower bounds) let
$$\mathcal{C}_d(D, S) \triangleq \mathcal{C}_{\Theta(D), 3d+\Theta(S), d}^{\mathrm{poly}(d), \mathrm{poly}(d)}.$$
This will comprise the family of ReLU network discriminators that we will focus on. We now show that if one could exhibit generators that can provably fool discriminators in $\mathcal{C}_d(D, S)$, then this would translate to average-case hardness against linear threshold circuits of depth $D$ and size $D$. Formally, we show the following:

**Theorem C.2.** *There is an absolute constant $c > 0$ for which the following holds. Fix sequences of parameters $D(m), S(m) \in \mathbb{N}$. Suppose there is an explicit[7] sequence of generators $G_m : \mathbb{R}^m \to \mathbb{R}^{d(m)}$ for $d(m) \geq cm \log m$ such that $W_{\mathcal{C}_{d(m)}(D(m), S(m))}(G_m(I_m), I_{d(m)}) \leq \epsilon(m)$ for some $\epsilon(m) \geq 1/\mathrm{poly}(m)$ and such that each output coordinate of $G_m$ is computable by a network in $\mathcal{F}^*$, then there exists a sequence of functions $h_{d(m)} : \{\pm 1\}^{d(m)} \to \{\pm 1\}$ in $\mathsf{NP}$ which are $(1/2+\epsilon(m)/2+m^{-\Omega(m)})$-average-case-hard with respect to some sequence of explicit distributions $\{\mathcal{D}_m\}$ for linear threshold circuits of depth $D(m)$ and size $S(m)$.*

*Remark* C.3. In particular, this shows that if we could exhibit explicit generators fooling all discriminators given by neural networks of polynomial Lipschitzness/bit complexity of depth $D(m)$ and size $O(d(m)^{1+\exp(-D(m)^{.99})})$, then by (2) we would get new average-case circuit lower bounds for $\mathsf{TC}^0$. In fact it was shown by (Chen & Tell, 2019) that such a result would imply $\mathsf{TC}^0 \neq \mathsf{NC}^1$, which would be a major breakthrough in complexity. This can be interpreted in one of two ways: 1) it would be extraordinarily difficult to show that a particular generative model truly fools all constant-depth, barely-superlinear-size ReLU network discriminators, or 2) gives a learning-theoretic motivation for trying to prove circuit lower bounds.

Regarding the proof of Theorem C.2, note that the statement is closely related to existing well-studied connections between hardness and randomness in the study of pseudorandom generators. In fact, readers familiar with this literature will observe that Theorem C.2 is the GAN analogue of the "easy" direction of the equivalence between hardness and randomness: an explicit pseudorandom generator that fools some class of functions implies average-case-hardness for that class.

In order to leverage this connection however, we need to formalize the link between GANs (over continuous domains) and pseudorandom generators (over discrete domains) in the next lemma. It

---

[6]Sometimes $\mathsf{TC}^0$ is defined with the gate set taken to consist of $\{\wedge, \vee, \neg\}$ and majority gates, though these two classes are equivalent up to polynomial overheads (Goldmann et al., 1992; Goldmann & Karpinski, 1998). Moreover, because a circuit of size $S$ and depth $D$ using the latter gate set is clearly implementable by a circuit of size $S$ and depth $D$ using the former gate set, so our lower bounds against the former gate set immediately translate to ones against the latter.

[7]By *explicit*, we mean that we are provided a way to evaluate these functions in polynomial time.

turns out that in the preceding sections we already developed most of the ingredients for establishing this connection.

**Lemma C.4.** *Suppose there is an explicit sequence of generators* $G_m : \mathbb{R}^m \to \mathbb{R}^{d(m)}$ *such that* $W_{\mathcal{C}_{d(m)}(D(m),S(m))}(G_m(I_m), I_{d(m)}) \leq \epsilon(m)$ *for some* $\epsilon(m) = 1/\mathrm{poly}(m)$ *and such that each output coordinate of* $G_m$ *is computable by a network in* $\mathcal{F}^*$. *Then there is an explicit sequence of pseudorandom generators* $G'_m : \{\pm 1\}^{n(m)} \to \{\pm 1\}^{d(m)}$ *for* $n(m) = \Theta(m \log m)$ *that* $2\epsilon(m)$-*fool linear threshold circuits of depth* $D(m)$ *and size* $S(m)$.

*Proof.* As in the proofs of the theorems from Section 3, the parameter $m$ will be clear from context, so we will drop $m$ from subscripts and parenthetical references.

Recall the function $h_\xi$ from Lemma 3.10; we will take $\xi = \epsilon/\mathrm{poly}(m)$. Also define $n \triangleq \Theta(\log(m/\epsilon))$ and recall from the proof of Lemma 3.8 the definition of the linear function $J : \mathbb{R}^{mn} \to \mathbb{R}^m$: for every $i \in [m]$, the $i$-th output coordinate of $J$ is the linear function which maps $x \in \mathbb{R}^{mn}$ to $\langle w_i, x + \mathbf{1} \rangle$, where $w_i$ is zero outside of indices $\{(i-1) \cdot n + 1, \ldots, i \cdot n\}$ and equal to the vector $(1/4, 1/8, \ldots, 1/2^{n+1})$ on those indices.

Given generator $G$ fooling $\mathcal{C}_d$, we will show that the Boolean function $G' : \{\pm 1\}^{mn} \to \{\pm 1\}^d$ given by

$$G' = h_\xi \circ G \circ J$$

is a pseudorandom generator that fools $\mathsf{TC}^0$ circuits. To that end, suppose there was a $\mathsf{TC}^0$ circuit $f : \{\pm 1\}^d \to \{\pm 1\}$ for which $|\mathbb{E}[f(G'(U_{mn}))] - \mathbb{E}[f(U_d)]| > 2\epsilon$. We will show that this implies the existence of a ReLU network $f' \in \mathcal{C}_d(D, S)$ for which $|\mathbb{E}[f'(G(I_m))] - \mathbb{E}[f'(I_d)]| > \epsilon$.

Our proof proceeds in three steps: argue that

1. $f \circ h_\xi \in \mathcal{C}_d(D, S)$

2. $\mathbb{E}[f(U_d)] \approx \mathbb{E}[f(h_\xi(I_d))]$

3. $\mathbb{E}[G'(U_{mn})] \approx \mathbb{E}[f(h_\xi(G(I_m)))]$

Note that 2 and 3, together with the fact that $f$ is a discriminator for $G'$, imply that $f' \triangleq f \circ h_\xi$ is a discriminator for $G$. 1 then ensures that this discriminator is a ReLU network with the right complexity bounds, yielding the desired contradiction.

To show step 1, we will show that $f$ can be computed by a network in $\mathcal{C}_{O(1),\mathrm{poly}(d),d}^{\mathrm{poly}(d),\mathrm{poly}(d)}$ and then apply Lemma 2.1 and Lemma 3.10. Suppose the threshold circuit computing $f$ has depth $D$, where $D$ is some constant. Recall from Lemma A.7 that we may assume, up to an additional blowup in size by $D$, that the constant-depth threshold circuit $C$ computing $f$ is comprised of layers $S_1, \ldots, S_D$ such that $S_i$ consists of all gates in $C$ for which any path from the inputs to the gate is of length $i$.

Let $k_i$ denote the number of gates in $S_i$ (where $k_D = 1$), and for each $j \in [k_i]$, suppose the linear threshold function computed by the $j$-th gate in $S_i$ is given by $\mathrm{sgn}(\langle w_{i,j}, \cdot \rangle - b_{i,j})$ for $w_{i,j} \in \mathbb{R}^{k_{i-1}}$. As each linear threshold takes at most $\mathrm{poly}(d)$ bits as input, we can assume without loss of generality that $b_{i,j}$ and the entries of $w_{i,j}$ lie in $\mathbb{R}_\tau$ for $\tau = \mathrm{poly}(d)$. For this $\tau$, note that for any $w \in \mathbb{R}_\tau^k, b \in \mathbb{R}_\tau, x \in \{\pm 1\}^k$,

$$\mathrm{sgn}(\langle w, x \rangle - b) = h_{\xi'}(\langle w, x \rangle - b),$$

for some $\xi' = 1/\mathrm{poly}(d)$, where $h_{1/\mathrm{poly}(d)}(\cdot)$ is the function defined in Lemma 3.10, and recall from the proof of Lemma 3.10 that it can be represented as a two-layer ReLU network via (15). For every $i \in [D]$, we can thus define two weight matrices $\mathbf{W}_i^{(1)} \in \mathbb{R}^{2k_i \times k_{i-1}}$ and $\mathbf{W}_i^{(2)} \in \mathbb{R}^{k_i \times 2k_i}$ by

$$\mathbf{W}_i^{(1)} = \frac{1}{\xi'} \cdot \begin{pmatrix} - & w_{i,1} & - \\ - & w_{i,1} & - \\ \vdots & \vdots & \vdots \\ - & w_{i,k_i} & - \\ - & w_{i,k_i} & - \end{pmatrix} \qquad \mathbf{W}_i^{(2)} = \begin{pmatrix} 1 & -1 & 0 & 0 & \cdots & 0 & 0 \\ 0 & 0 & 1 & -1 & \cdots & 0 & 0 \\ \vdots & \vdots & \vdots & \vdots & \ddots & \vdots & \vdots \\ 0 & 0 & 0 & 0 & \cdots & 1 & -1 \end{pmatrix}$$

and biases $b_i^{(1)} \in \mathbb{R}^{2k_i}$ and $b_i^{(2)} \in \mathbb{R}^{k_i}$ by

$$b_i^{(1)} = (1, -1, 1, -1, \ldots, 1, -1) \qquad b_i^{(2)} = (-1, \ldots, -1)$$

so that for all $x \in \{\pm 1\}^d$,

$$f(x) = \mathbf{W}_D^{(2)} \phi \left( \mathbf{W}_D^{(1)} \phi \left( \cdots \phi \left( \mathbf{W}_1^{(2)} \phi \left( \mathbf{W}_1^{(1)} x + b_1^{(1)} \right) + b_1^{(2)} \right) \cdots \right) + b_D^{(1)} \right) + b_D^{(2)} \qquad (17)$$

The entries of the weight matrices and bias vectors are clearly in $\mathbb{R}_{\text{poly}(d)}$, and because each $h_{\xi'}$ is $\text{poly}(d)$-Lipschitz and there are $D = O(1)$ layers in the circuit, the function in (17) is $\text{poly}(d)$-Lipschitz as a function over $\mathbb{R}^d$. The size and depth of the network are within a constant factor of the size $S$ and depth $D$ of the circuit. Lemma 2.1 and Lemma 3.10 then imply that $f \circ h_\xi$ has depth $\Theta(D)$ and size $3d + \Theta(S)$, as well as Lipshitzness and bit complexity polynomial in $m$ because $\epsilon \geq 1/\text{poly}(m)$ so that $\xi \geq 1/\text{poly}(m)$. Therefore, $f \circ h_\xi \in \mathcal{C}_d(D, S)$.

To show step 2, recall from Lemma 3.11 and Remark 3.12 that $W_1(U_m, h_\xi(I_m)) \leq \epsilon/\text{poly}(m)$. Recalling that $f$ is $\text{poly}(d) = \text{poly}(m)$-Lipschitz, we obtain the desired inequality $|\mathbb{E}[f(U_d)] - \mathbb{E}[f(h_\xi(I_d))]| \leq \epsilon/2$. Here the factor of $1/2$ is an arbitrary small constant coming from taking the $\text{poly}(m)$ in the definition of $\xi$ sufficiently large.

Finally, to show step 3, recall by Fact 3.7 that $W_1(J(U_{mn}), I_m) \leq \epsilon^2/\text{poly}(m)$ by our choice of $n = \Theta(\log(m/\epsilon))$ (the $\epsilon^2$ comes from taking the constant factor in the definition of $n$ sufficiently large). By applying Fact A.2 to $f \circ h_\xi$ and $G$, we know that the composition $f \circ h_\xi \circ G$ is $\text{poly}(m)/\epsilon$-Lipschitz. It follows that $|\mathbb{E}[G'(U_{mn})] - \mathbb{E}[f(h_\xi(G(I_m)))]| \leq \epsilon/2$. The factor of $1/2$ is an arbitrary small constant coming from taking the constant factor in the definition of $n$ sufficiently large.

Putting everything together, we conclude by triangle inequality that

$$|\mathbb{E}[f(G(I_m))] - \mathbb{E}[f(I_d)]| > \epsilon,$$

a contradiction. $\qquad \square$

The following lemma gives the standard transformation from pseudorandom generators to average-case hardness. We include a proof for completeness.

**Lemma C.5** (Prop. 5 of (Viola, 2009)). *Suppose the sequence of functions $G_m : \{\pm 1\}^m \to \{\pm 1\}^{d(m)}$ $\epsilon(m)$-fools a class of Boolean functions $\mathcal{F}$. Define the function $h_{d(m)} : \{\pm 1\}^{d(m)} \to \{\pm 1\}$ by*

$$h_{d(m)}(x) = \begin{cases} 1 & \text{exists } y \in \{\pm 1\}^m \text{ such that } G(y) = x \\ -1 & \text{otherwise} \end{cases}.$$

*Let $\mathcal{D}_{d(m)}$ be the distribution over $\{\pm 1\}^{d(m)}$ given by the uniform mixture between $U_{d(m)}$ and $G(U_m)$.*

*Then the sequence of functions $\{h_{d(m)}\}$ is $(1/2 + \epsilon'(m))$-average-case-hard for $\mathcal{F}$ with respect to $\{\mathcal{D}_{d(m)}\}$ for $\epsilon'(m) = \epsilon(m)/4 + 2^{m-d(m)-1}$.*

*Proof.* As usual, we will omit most subscripts/parentheses referring to the parameter $m$. Let $f : \{\pm 1\}^d \to \{\pm 1\}$ be any function in $\mathcal{F}$. Then

$$\mathbb{P}[f(\mathcal{D}) = h_d(\mathcal{D})] = \frac{1}{2} \mathbb{P}[f(U_d) = h_d(U_d)] + \frac{1}{2} \mathbb{P}[f(G(U_m)) = h_d(G(U_m))]$$

$$\leq \frac{1}{2} \left( \mathbb{P}[f(U_d) = 0] + \mathbb{P}[h_d(U_d) = 1] \right) + \frac{1}{2} \mathbb{P}[f(G(U_m)) = 1]$$

$$\leq \frac{1}{2} \left( \mathbb{P}[f(U_d) = 0] + 2^{m-d} \right) + \frac{1}{2} \mathbb{P}[f(G(U_m)) = 1]$$

$$\leq \frac{1}{2} \left( \mathbb{P}[f(U_d) = 0] + 2^{m-d} \right) + \frac{1}{2} \left( \mathbb{P}[f(U_d) = 1] + \epsilon/2 \right)$$

$$= \frac{1}{2} + \frac{\epsilon}{4} + 2^{m-d-1},$$

where in the second step we used a union bound and the fact that $h(G(U_m))$ is deterministically 1 by construction, in the third step we used the fact that $\mathbb{P}[h_d(U_d)] \leq 2^{m-d}$ because there are at most $2^m$ elements in the range of $G$, and in the fourth step we used the fact that $G$ $\epsilon$-fools functions in $\mathcal{F}$. $\qquad \square$

We are now ready to prove Theorem C.2.

*Proof of Theorem C.2.* By Lemma C.4, we can construct out of the generators $G_m$ an explicit sequence of pseudorandom generators that stretch $\Theta(m \log m)$ bits to $d(m) \geq c \cdot m \log m$ bits and $2\epsilon(m)$-fool linear threshold circuits of size $S(m)$ and depth $D(m)$. The theorem follows upon substituting this into Lemma C.5, which implies $(1/2 + \epsilon'(m))$-average-case-hardness for such circuits with respect to the explicit distributions $\mathcal{D}_{d(m)}$ defined in Lemma C.5, where $\epsilon'(m) = \epsilon(m)/2 + 2^{\Theta(m \log m) - d(m) - 1} = \epsilon(m)/2 + m^{-\Omega(m)}$, provided the absolute constant $c$ is sufficiently large.

Finally, note that the average-case-hard functions $h_{d(m)}$ we get from Lemma C.5 are in NP because given an input $x$ and a certificate $y$, one can easily verify whether $G(y) = x$. $\square$

