# OpenReview forum: "Minimax Optimality (Probably) Doesn't Imply Distribution Learning for GANs"
_ICLR.cc/2022/Conference — ICLR 2022 Poster_

### Official Review · Reviewer_h3Jv · 2021-10-23

**Correctness:** 3
**Technical Novelty And Significance:** 4
**Empirical Novelty And Significance:** 3
**Recommendation:** 6
**Confidence:** 3

**Main Review:**

Major Comments:

- The potential impact of this paper is strong: by revealing that the class of poly-sized discriminator may not be rich enough to achieve distributional learning, the paper motivate us to reconsider if GAN is more suitable for feature learning rather than distributional learning. Given the successful empirical performance of GAN, one may also reconsider if Wasserstein distance is a reasonable objective function to use.

- The assumptions imposed by the paper such as poly-sized neural network discriminators and diverse target distribution seems natural.

- To the best of my knowledge, it is novel to apply pseudorandom generators theory to study GAN.

- The numerical part is simple but enough to support the validity of the theory.

Minor Comments (several typos in the paper):

- Definition 2: $\mathbb E_{x\sim q}(f(y))$
should be replaced by
$\mathbb{E}_{x\sim q} (f(x))$?

- Eq (1): $\mathcal D_{d(m)}$
should be replaced by
$\mathcal D^{\ast}_{d(m)}$?

- Last sentence in the proof sketch of Theorem 3.2: we get that there exist a threshold $t\in \mathbb{R}_{\mathrm{poly}(d)}$ for which $|\mathbb{E}[\mathcal{M}_\tau(G(U_m))] - \mathbb{E}[\mathcal{M}_\tau(U_d)]|,...$, I suspect the inequility for $|\mathbb{E}[\mathcal{M}_\tau(G(U_m))] - \mathbb{E}[\mathcal{M}_\tau(U_d)]|$ is incomplete here.

**Summary Of The Paper:**

The paper leverages the techniques in cryptographic to prove a surprising result that a uniform small error against against all poly-size neural network discriminators can not guarantee a small error in type-1 Wasserstein distance.

**Summary Of The Review:**

The paper provides a new perspective on the important problem if GAN is able to achieve distributional learning. In general, the paper is well-written, and to the best of my knowledge, the proposed methodology is novel to this type of problem. Thus I recommend acceptance.

---

> ### Author Response · Authors · 2021-11-17
> **Thanks for the review!**
>
> Thanks for the positive feedback, the major comments listed are a great overview of the main contributions of our work! We will make sure to incorporate the typo fixes into the upcoming revision.

---

### Official Review · Reviewer_yN8S · 2021-10-28

**Correctness:** 3
**Technical Novelty And Significance:** 3
**Empirical Novelty And Significance:** 1
**Recommendation:** 3
**Confidence:** 3

**Main Review:**

The paper reduce the problem of "whether there exists a bad solution of GAN objective" to the existence of local pseudorandom generators. This is an interesting and novel idea. Then the paper shows that under some artificial case, the min-max optimality cannot imply that the GAN learns the distribution. The paper is highly theoretical.

My major concern is the significance of the result. It seems to be well-known and intuitive that fooling a weak class of discriminators does not imply exactly learning the target distribution.

Besides, I would like to provide some suggestions for the paper writing.

1. In Theorem 1.1, the $\gamma_m$ is not defined. This makes the theorem hard to understand.

2. Definition 5 is extremely difficult for people who are unfamiliar with cryptography to understand. What do "uniformly random k-uniform hyper-graph" and "k-ary predicate" mean? I also cannot understand what is "circuits in P/poly".

3. In Lemma 2.4, the authors should indicates the failure probability, that is, with probability at least $1-\delta$ the result holds. Otherwise the lemma just looks erronous.

4. There are a lot of Lemmas that are spread across the paper that make the paper looks somewhat broken and also increase the burden for readers to understand. For example, Lemma 2.1 is only used in the Proof of Theorem 3.6. Lemma 2.3 looks immediate from the definition and is never used in the paper. The authors may not want to present those Lemmas in the paper.

**Summary Of The Paper:**

The paper shows that if Goldreich’s PRG is able to fool all Boolean circuits of polynomial size, then we can construct a poly size ReLU network as a generator that outputs a distribution that has a constant $W_1$ distance to the target distribution but all poly-size ReLU networks cannot discriminate between the two distributions.


**Summary Of The Review:**

The paper is highly theoretical. My major concern is the significance of the result. It seems to be well-known and intuitive that fooling a weak class of discriminators does not imply exactly learning the target distribution.

---

> ### Author Response · Authors · 2021-11-17
> **Thanks for the review, here we clarify several points that were raised.**
>
> The primary criticism that was made was that "It seems to be well-known and intuitive that fooling a weak class of discriminators does not imply exactly learning the target distribution."
> - Note that **the statement “fooling a weak class of discriminators does not imply exactly learning the target distribution” is not true in general**. For instance, if the generator is a linear map, as long as it fools a very simple family of discriminators, namely quadratic functions, then it must learn any target distribution which is a linear pushforward of a Gaussian. In contrast, our hardness result comes from the fact that the generator is nonlinear (i.e. constant-depth relu network), enabling it to fool a large class of target distributions with respect to a strong family of discriminators.
> - Moreover, note that **the class of discriminators we consider is not weak**. In fact, the class of discriminators we consider is essentially the *strongest possible* that one could use in practice, because the set of polynomial-size neural networks by definition captures the largest family of discriminators one could hope to evaluate efficiently. To reiterate, the main result is to show that there are very simple generators which can fool this very general class of discriminators but which yield distributions that are statistically far from the target distribution.
> - Finally, note that our result is **not about hardness of “exactly” learning**. We show that you can fool all poly-size discriminators to negligibly small error and still be off from the target distribution even by a *constant distance* in Wasserstein.
>
> We next address the other suggestions made in the review:
> 1) *$\gamma_m$ is not defined*: In the upcoming revision, we will move the definition of $\gamma_m$ from the preliminaries to before the statement of our main result.
> 2) *Terminology difficult to understand*: The notions of P/Poly, random hypergraphs, and k-ary predicates are used frequently in complexity theory and cryptography, see e.g. the survey https://eprint.iacr.org/2015/165.pdf. That said, we will make sure to include a more thorough introduction of these topics in the supplementary material in the upcoming revision.
> 3) *Lemma 2.4*: Yes, there should be a “with high probability” in the statement.
> 4) *Lemma 2.1, many lemmas spread across paper*: We use Lemma 2.1 throughout the proofs of our results, but because we deferred most of the proofs in the main body to the supplementary material because of space constraints, this is not apparent from the main body. We will make sure to reorganize the material and add explanatory text in the upcoming revision to address these concerns.
>
> **To summarize:** while our results are theoretical, we are attacking a practically important question about the efficacy of approximating distributional distances using neural networks. We approached this from the theoretical standpoint of cryptography and complexity theory and showed an impossibility result. As described earlier in the rebuttal, the class of discriminators we consider is not weak but essentially the strongest possible.
>
> Furthermore, although it is perhaps believed by some practitioners that GANs do not actually perform distribution learning in a traditional statistical sense, ours is the first work to put this belief on rigorous foundations. We also remark that, as mentioned in our related works section, there are numerous lines of theoretical work attempting to demonstrate that minimax optimality of GANs does imply distribution learning, under restrictive assumptions. One can view our result as a “no-go” theorem, stating that these results do not extend to situations of practical interest. *We are not aware of any prior work establishing a guarantee of this nature.*

---

> > ### Author Response · Authors · 2021-11-22
> > **Let us know if you have any questions!**
> >
> > As today is the last day for paper revisions, and given the rather strong nature of the review, it would be good to know if we have adequately addressed your concerns. Respectfully, we disagree with the criticism of the novelty of our results, and it seems that the other reviewers agree that our contributions are new and interesting and raise intriguing new questions regarding what one can rigorously prove about GANs.

---

### Official Review · Reviewer_qEsw · 2021-11-02

**Correctness:** 3
**Technical Novelty And Significance:** 3
**Empirical Novelty And Significance:** 3
**Recommendation:** 6
**Confidence:** 4

**Main Review:**

The results are novel, interesting, and non-trivial. The construction draws connections between GANs and local PRGs which is interesting.

The authors interpret the result of the paper as saying "minimax optimality does not imply distribution learning". This is by itself a rather inaccurate statement: distribution learning is naturally defined based on a measure of success. The authors have shown that minimax optimality does not imply learning w.r.t. Wasserstein distance (under some assumptions). The choice of Wasserstein is natural but somewhat arbitrary. For example, if we change the Wasserstein to another IPM (e.g,. one which is directly defined based on set of neural nets rather than Lipschitz functions) then the result would be false. In fact part of the success of GANs in applications like image generation can be attributed to the fact that they don't really optimize the usual notions of distance/divergence between distributions. This has been discussed to some extent in the conclusions of the paper, but I still think the message of the paper in other parts of the paper can be misleading.

I think the presentation of the paper can be significantly improved. More specifically, some of the notations are hard to follow. Moreover, some background on PRGs are missing in the main text, making it hard to follow the paper for those who are not familiar with them (I suppose this is the case for most of the audience of this ICLR submission)

My understanding is that the authors use random networks as their generators, and this enables to have generators that receive discreet seed distributions but the distribution of their output is continuous. Therefore my understanding is that one of the main reductions of the paper does not go through with the use of deterministic (e.g, ReLU) nets. Please elaborate. If this is the case, then it should be mentioned clearly, since most generators used in practice adopt deterministic networks.

The experiments section is rather weak. For one, the target distribution is chosen to be discreet whereas in usual applications the target distribution is rather continuous (like images).

=================
More comments

+ In Theorem 1, $\gamma_m$ is not defined before.
+ throughout the intro (e.g., theorem 1), we see "polynomial discriminator" and "polynomial generator"; it would be helpful if you mention the parameter(s) w.r.t. which we are taking about in each case
+ would be helpful to define negl function in the main text since it's been used in the main theorem statements
+ In Thm 3.1 we see that the size of the parameters as well as W_{F*} depend on \epsilon(m). This makes it hard to evaluate the strength of the bound. In other words, the generator is also getting more complicated as m grows. Can you demonstrate the power of your bound by choosing a good \epsilon(m)?
+ In equation (1) we see $D_{d(m)}$. Should this be $f(D^*_{d(m)})$?



**Summary Of The Paper:**

The papers offers strong evidence that even if (population) minimax optimality is satisfied for a GAN the generator may not have learned the distribution with respect to the Wasserstein distance. More specifically, the authors show that there exits distributions that:

+ can be generated by simple generators nets (variants of randomized ReLU nets with constant depth and polynomial size and Lipschitzness) that are fed with simple seed distributions (e.g., uniform on a cube)
+ are discrete and far (in Wasserstein distance) from any "diverse" distribution
+ cannot be distinguished from (a simple) diverse target distribution using any polynomial discriminator network (polynomial size/depth/Lipschitzness)

The proof is based on a cryptographic assumption, namely the existence of a local pseudo-random generators (they use a specific one proposed by Goldreich).

**Summary Of The Review:**

This is a solid work and proves an interesting result. The presentation can be improved, and in general the paper is not easy to read. The experiments are quite weak.

---

> ### Author Response · Authors · 2021-11-17
> **Thanks for the review and for raising interesting questions! Here we clarify some important points.**
>
> We begin by addressing the four points in the main review:
> 1) *Wasserstein and distribution learning*: Thanks for raising the important question of how to quantify distribution learning! We used Wasserstein distance as our proxy for distribution learning as it is the most standard notion for statistical distance in this literature; indeed, practically all prior works that sought to understand when minimax optimality implies statistical closeness consider this distance (see Section 1.1). That said, we agree it is essential to understand whether there are more sophisticated notions of distribution learning than the traditional metrics/divergences that better capture what GANs actually accomplish. We see our work as a necessary first step towards even formally articulating what these new notions should be. *Alternatively, our result can be viewed as "closeness in the Wasserstein  GANs does not imply closeness in the conventional Wasserstein distance" *
> 2) *Presentation and background on PRGs*: We will make sure to fill in additional background on PRGs and other complexity-theory/crypto notions in the supplementary material in the upcoming revision. We will also make sure to render the notation more transparent in the revision.
> 3) *Random networks, seed/output distributions?*: First note that our main results are about generators which receive *continuous* seed distribution (namely Gaussian) and output a *continuous* distribution.
> As for whether the generators we use are random networks, this is not the case. We emphasize that the generators we construct are all *deterministic functions* that compute fixed ReLU networks. The cryptographic assumption we use merely says that most choices of architecture for the PRG ($H$ in Assumption 1) satisfy the pseudorandomness properties we need.
> 4) *Experiments section*: Our contribution is primarily theoretical. As reviewers v7TX and h3Jv remark, our experiments were for illustrative purposes: the instance we evaluated empirically is arguably the simplest example of a distribution which is far in Wasserstein from some target distribution but can fool neural network discriminators of bounded complexity.
>
> Finally, we briefly address the points raised in "More comments." Regarding $\gamma_m$, the "polynomial" terminology, and "negl", we will incorporate these fixes into the upcoming revision. As for (1), it indeed should be $f(D^*_{d(m)})$, thanks for the catch! Finally, regarding $\epsilon(m)$, one can simply take it to scale inversely in the Lipschitzness of the network generating the target distribution. We will also clarify this in the upcoming revision.

---

> > ### Comment · Reviewer_qEsw · 2021-11-22
> > **updated the score**
> >
> > Thanks for the clarifications (including the one about random networks). I updated my score accordingly.

---

### Official Review · Reviewer_DfoH · 2021-11-02

**Correctness:** 3
**Technical Novelty And Significance:** 3
**Empirical Novelty And Significance:** Not applicable
**Recommendation:** 6
**Confidence:** 3

**Main Review:**

The paper is interesting. However, I have the following questions about the presentation and its significance.

1. The assumption on the diversity of the data distribution is not discussed in detail, especially the one used in the main Theorem. The authors only say it is a large family of distributions while it is unclear and not so intuitive for readers.

2. How large is the $\epsilon(m) poly(m)$ in Theorem 3.1? Is it meaningful in practice? Does the result provide any insight on training GANs since the estimation of the Wasserstein-1 metric is monitored during training?

3. As discussed in the conclusion, the paper only proves the existence of a "bad" generator while optimizing the GAN objective does not always lead to the worst case. The authors claim "MINIMAX OPTIMALITY $\textbf{(PROBABLY)}$ DOESN’T IMPLY DISTRIBUTION LEARNING FOR GANS" while the paper does not discuss how likely we really meet this in practice, which limits the significance of the paper.

4. There is a gap between the experiments and theory. The authors train 4 different discriminators, which may not cover all of the possible ones. It would be better if the authors can conduct an example where the generator can cheat all of the discriminators.

**Summary Of The Paper:**

This paper studies the problem of learning generative adversarial networks using a ploy-size ReLU generator and discriminator under the standard Wasserstein-1 metric. The main result is that there exists a "bad" generator that can cheat all discriminators under the estimation of the Wasserstein-1 metric while being far from the data distribution under the true Wasserstein-1 metric. The proof relies on two assumptions. The first one is on the diversity of the target data distribution. The second one is a standard assumption in cryptography as claimed by this paper while I'm not familiar with cryptography. The results explicitly consider the computation complexity of the model and may show the learnability of the GAN model in some sense.

**Summary Of The Review:**

Overall, I think this is an interesting paper and currently, I tend to accept it if all of the concerns are well addressed.

---

> ### Author Response · Authors · 2021-11-17
> **Thanks for the review and for the helpful questions!**
>
> Here we address the four main questions posed:
> 1) *Diversity assumption*: The assumption of diversity is very mild and would be satisfied by any real-world distribution; all it’s saying is that the distribution isn’t just concentrated around a small collection of points. Note that in the main body (see paragraph after Lemma 2.3), we give several examples, for instance Lemma 2.4 shows that almost all expansive leakyrelu networks yield diverse distributions. We will make sure to elaborate on this assumption in the upcoming revision.
> 2) *$\epsilon(m)\text{poly}(m)$*: The extra $\text{poly}(m)$ factor merely comes from the $\text{poly}(m)$ Lipschitz-ness/bit-complexity of the generative model $H_m$ that generates the target distribution, and $\epsilon(m)$ is an arbitrary parameter that one can choose, as long as it is not exponentially small. In particular, if one wanted $W_{F*}$ distance small in bullet point 2 of our guarantee, one just needs to take $\epsilon(m)$ scaling inversely with the Lipschitzness of $H_m$. This would only incur a $\log$(Lipschitzness) overhead in the size of $G_m$, so the guarantee is definitely meaningful in practice. We thank the reviewer for the question and will incorporate this part of our response into the upcoming revision to make the guarantee in Theorem 3.1 more transparent.
> Additionally, the question about training GANs is very interesting and is one of the primary open questions we raise in our conclusion section. Our result only guarantees the existence of bad global optima, so it would be interesting to study whether the training dynamics can help avoid these bad optima.
> 3) *Practical significance*: Note that in the second paragraph of the conclusion, we do say “Of course, it is quite unlikely that we will ever encounter such a generator through natural GAN training. One way to circumvent our lower bound is to argue that the training dynamics of the generator may have some regularization effect...”
> We would like to emphasize that the theory for distribution learning for GANs is very much in its nascent stages, and the field is still far from a rigorous understanding for why (and whether) it is true that GANs efficiently achieve distribution learning. Our main contributions are to take a **mathematically principled first step towards clarifying what one can and cannot hope for in such a theory**, and to elucidate **powerful new connections to cryptography and complexity theory** that  are evidently crucial for making further progress on the theoretical front for understanding GANs.
> 4) *Experiments*: Our theorem already proves under standard cryptographic assumptions that this particular instance cheats all polynomial-size discriminators, and the experiments are primarily for illustrative purposes (as reviewers v7TX and h3Jv also note). It is unclear what it means to experimentally demonstrate that a generator can cheat *all* discriminators as this would require an exhaustive enumeration over exponentially many possible functions.

---

> > ### Comment · Reviewer_DfoH · 2021-11-24
> > **Thanks for the feedback and I keep my score**
> >
> > Thank you for the detailed feedback and my major concerns are addressed. I believe this is a theoretical attempt that is worth being published in ICLR and I keep my recommendation. However, I still suggest that the authors clarify the gap between the experiments and theory in Sec. 4 somewhere. The current claim "To empirically demonstrate the existence of a constant depth generator that can fool polynomially bounded discriminators, " is somewhat misleading.

---

> > > ### Author Response · Authors · 2021-11-28
> > > **Thanks!**
> > >
> > > We really appreciate the positive feedback and will make sure to reword our framing of the experimental results accordingly!

---

### Official Review · Reviewer_v7TX · 2021-11-02

**Correctness:** 4
**Technical Novelty And Significance:** 4
**Empirical Novelty And Significance:** 1
**Recommendation:** 6
**Confidence:** 4

**Main Review:**

The paper is well organized. Although there are many technical details, which are relatively hard to follow every bit, it is due to the rigor and complexity of the theory. I like the flow of the paper, especially building the theory from easier discrete cases to continuous generalization. Yet some improvement can be made in Section 3.2 and Section 3.3. For example, the connection between Section 3.2 and Section 3.1 is somewhat vague, and some high level idea of extending binary output to continuous output is helpful appearing before Theorem 3.6.

The experiments are for illustrative purpose. However, I find it a bit confusing. Does figure 1 report the training loss or testing loss? If I understand correctly, we should achieve approximately zero training loss, while the Wasserstein distance between the data distribution and the generated distribution shows a nonzero gap. In figure 1, the Wasserstein distance is claimed to be large, without numerical verification. By the way, I am curious how is the loss $\mathbb{E}[-\log (D(X))] + \dots$ is computed.

**Summary Of The Paper:**

The paper provides an interesting negative result on that polynomial sized discriminator lacks sufficient discriminative power to distinguish the data distribution and generated distribution by a constant depth generator. The argument rigorously indicates that a small neural IPM over the discriminator network can still yield a large Wasserstein distance.

**Summary Of The Review:**

The theory in the paper utilizes polynomially-sized Boolean circuit theory, which is an interesting connection. The paper does not have obvious contribution to practically trained GAN models, however, this hardness result provides revealing insights of GANs. In fact, the paper opens some directions to investigate and should be highly important to GANs. For example, if we change the architecture of the generator, does the discriminator in the paper still lack power? Maybe an easier (somewhat orthogonal) question is if the generator network is powerful in representing the data distribution in Wasserstein distance, does there exists good choice of discriminator (poly size for example) to guarantee the distribution recovery in Wasserstein distance. Overall, I am positive on the paper.

---

> ### Author Response · Authors · 2021-11-18
> **Thanks for the review and the interesting questions!**
>
> We appreciate the positive feedback on the technical writing and on the significance of our results to the theory of GANs. We very much agree that the utility of our negative result is primarily in suggesting new directions for understanding how GANs perform distribution learning.
>
> Here we address three main points raised in the review, namely 1) the experimental results, 2) whether modifying the architecture can circumvent cryptographic hardness, and 3) what happens if the generator family is sufficiently expressive to capture the data distribution in Wasserstein.
>
> 1) *Experimental results*: There is no need to numerically verify that the Wasserstein is large, as this holds trivially: we are comparing the uniform distribution over $\{\pm 1\}^{200}$ to the output of Goldreich’s PRG, which is only supported on $2^{50}$ points, so the Wasserstein distance must be large (see Lemma 2.3 and also Lemma A.13 in the supplement). What is being verified in Figure 1 is that the discriminators we trained are unable to distinguish between these two distributions, so in Figure 1 we have plotted the test loss. The test loss is simply computed by an empirical average over samples. We will add text to the upcoming revision clarifying these points.
> 2) *Architecture*: This is a great question! We do think this must be studied *in conjunction* with the training dynamics however: in isolation, modifying the architecture of the generator is most likely insufficient to avoid cryptographic obstacles because local PRGs are so simple that constraining the architecture to avoid these functions might render the generator class insufficiently expressive.
> 3) *Powerful generator classes*: Note that the results we present are precisely of the flavor you describe. Namely, we give instances where the data can be *perfectly* expressed by a simple generative model ($H_m$ in Theorem 3.1), but for which even fooling *all poly-size discriminators* does not suffice to recover the true distribution. We will clarify this point in the revision.

---

> > ### Comment · Reviewer_v7TX · 2021-11-23
> > **Thanks for the clarification on experiments and discussions of comments**
> >
> > I am happy to keep my positive review, and believe this is an interesting theoretical finding on polynomially sized GANs.

---

### Author Response · Authors · 2021-11-18
**New revision uploaded**

We thank the reviewers again for their helpful suggestions which we have incorporated into the new revision! Here we summarize the primary changes to the pdf:
- Additional prose throughout Section 3 giving more high-level intuition for how the different lemmas and sections fit together
- Reworded Section 2.2 to be more accessible to readers unfamiliar with cryptography/complexity theory
- Additional discussion after Theorem 1.1 clarifying confusions that arose in the reviews, e.g.
     - Our constructions are all families of *deterministic* functions
     - The seed and output of our GANs are both continuous
     - The discriminator class we consider is the strongest possible one that can be implemented in practice
     - The target distribution can be perfectly represented by some other generator
     - "Polynomial" refers to the dependence on the parameter $m$
- Elaborated in Section 2.3 on the generality and practical relevance of our "diversity" assumption
- Discussion after Theorem 3.1 about $\epsilon(m)\text{poly}(m)$ and how to pick $\epsilon(m)$
- Negligible functions defined in main text (footnote 4 on P. 5)
- Clarification about what is being shown in Figure 1
- Miscellaneous typo fixes suggested in the reviews

---

### Decision · Program_Chairs · 2022-01-20

**Decision:**

Accept (Poster)

**Comment:**

The provides a complexity theoretic look at GANs. The exposition is multi-disciplinary, and in my personal opinion, it is an interesting look at the GANs in the context of random number generators.